# Chemical structure-guided design of dynapyrazoles, cell-permeable dynein inhibitors with a unique mode of action

Jonathan B Steinman[1], Cristina C Santarossa[1], Rand M Miller[1], Lola S Yu[1], Anna S Serpinskaya[2], Hideki Furukawa[3], Sachie Morimoto[3], Yuta Tanaka[3], Mitsuyoshi Nishitani[4], Moriteru Asano[3], Ruta Zalyte[5], Alison E Ondrus[6], Alex G Johnson[7], Fan Ye[8], Maxence V Nachury[8], Yoshiyuki Fukase[3], Kazuyoshi Aso[3], Michael A Foley[3], Vladimir I Gelfand[2], James K Chen[7], Andrew P Carter[5], Tarun M Kapoor[1]*

[1]Laboratory of Chemistry and Cell Biology, Rockefeller University, New York, United States; [2]Department of Cell and Molecular Biology, Feinberg School of Medicine, Northwestern University, Chicago, United States; [3]Tri-Institutitional Therapeutics Discovery Institute, New York, United States; [4]Pharmaceutical Research Division, Takeda Pharmaceuticals Ltd, Kanagawa, Japan; [5]Medical Research Council Laboratory of Molecular Biology, Cambridge, United Kingdom; [6]Division of Chemistry and Chemical Engineering, California Institute of Technology, Pasadena, United States; [7]Department of Chemical and Systems Biology, Stanford University School of Medicine, Stanford, United States; [8]Department of Molecular and Cellular Physiology, Stanford University, Stanford, United States

*For correspondence: kapoor@rockefeller.edu

**Competing interests:** The authors declare that no competing interests exist.

**Abstract** Cytoplasmic dyneins are motor proteins in the AAA+ superfamily that transport cellular cargos toward microtubule minus-ends. Recently, ciliobrevins were reported as selective cell-permeable inhibitors of cytoplasmic dyneins. As is often true for first-in-class inhibitors, the use of ciliobrevins has in part been limited by low potency. Moreover, suboptimal chemical properties, such as the potential to isomerize, have hindered efforts to improve ciliobrevins. Here, we characterized the structure of ciliobrevins and designed conformationally constrained isosteres. These studies identified dynapyrazoles, inhibitors more potent than ciliobrevins. At single-digit micromolar concentrations dynapyrazoles block intraflagellar transport in the cilium and lysosome motility in the cytoplasm, processes that depend on cytoplasmic dyneins. Further, we find that while ciliobrevins inhibit both dynein's microtubule-stimulated and basal ATPase activity, dynapyrazoles strongly block only microtubule-stimulated activity. Together, our studies suggest that chemical-structure-based analyses can lead to inhibitors with improved properties and distinct modes of inhibition.

## Introduction

The AAA+ (ATPases Associated with diverse cellular Activities) superfamily is comprised of ~100 proteins in humans (*Erzberger and Berger, 2006*; Human AAA+ protein count was obtained as follows: in supfam.org, search was performed for "Extended AAA-ATPase domain" and refined for proteins within the human genome). These ATPases are essential for many cellular processes, including DNA replication, proteostasis, membrane remodeling, and cytoskeletal organization (*Hanson and Whiteheart, 2005*). Extensive cell biological and biochemical studies have revealed that these enzymes

couple ATP hydrolysis to substrate remodeling and directional transport, processes that can occur on the timescale of seconds or minutes (*Hanson and Whiteheart, 2005*). Thus, small molecule inhibitors that can modulate AAA+ activity on similarly fast timescales are likely to be valuable tools to probe their cellular functions (*Lampson and Kapoor, 2006*). Valosin-containing protein (*Deshaies, 2014*) and dynein (*Firestone et al., 2012*) are the only two human enzymes in this large superfamily for which well-characterized small molecule antagonists have been reported.

Dyneins are microtubule-based motor proteins in the AAA+ family that have been divided into two classes, axonemal and cytoplasmic. Axonemal dyneins are required for the beating of flagella. Cytoplasmic dyneins, of which there are two isoforms (hereafter, dynein 1 and 2), are present in metazoan cells and are required for a wide range of cellular processes (*Vale, 2003*; *Allan, 2011*; *Vallee et al., 2004*). Transport of cargo along microtubules requires a balance of forces directed toward opposite ends of the filament. While multiple motor proteins in the kinesin family provide the plus-end directed force to drive this motion, their activity in many contexts is opposed by one of two cytoplasmic dyneins, the primary motor proteins transporting cargos toward the minus-end of microtubules (*Vale, 2003*). Dynein 1 has many functions in the cytoplasm, where it moves diverse cargoes ranging from mRNA molecules to whole organelles. In contrast, dynein 2's functions are restricted to cilia and flagella. The primary cilium is an antenna-like organelle that protrudes from the cell surface. Bidirectional transport in the cilium, known as intraflagellar transport, is required for Hedgehog signaling, a developmental pathway (*Goetz and Anderson, 2010*). Cargos of both dynein isoforms can move at rates of >1 µm/s in cells (*Allan, 2011*; *Mijalkovic et al., 2017*) and therefore, fast-acting, reversible chemical inhibitors are likely to be useful probes for dynamic dynein-dependent cellular processes.

Ciliobrevins were recently reported as the first selective, cell-permeable probes of dynein (*Firestone et al., 2012*). Although other chemical antagonists of dynein have contributed to understanding the biochemistry of dynein, their use in cell biology has been limited because they either have limited cell permeability (e.g. vanadate) or are non-selective in cells (e.g. EHNA) (*Kobayashi et al., 1978*; *Gibbons et al., 1978*; *Bouchard et al., 1981*). Ciliobrevins were discovered as inhibitors of Hedgehog signaling and shown to block cytoplasmic dynein 1- and 2-dependent transport in cells (*Firestone et al., 2012*; *Hyman et al., 2009*). Ciliobrevins have been used as tools to examine the role of dynein in a number of processes, including formation of the immunological synapse, transport of signaling proteins in the primary cilium, axonal transport of transcription factors, and axon extension and branching in cultured neurons (*Sainath and Gallo, 2015*; *Liu et al., 2013*; *Yi et al., 2013*; *Ye et al., 2013*). However, the use of ciliobrevins has been limited by their low potency and suboptimal chemical properties (*Roossien et al., 2015*), which is often noted for first-in-class compounds identified through high-throughput screening. Complete inhibition of dynein by ciliobrevins can require high doses (50–100 µM) and selective protein target inhibition can be difficult to achieve at such concentrations.

The ciliobrevins are based on a benzoylacrylonitrile-substituted quinazolinone scaffold (*Figure 1A*). This type of acrylonitrile has the potential to react with nucleophiles, and instability of ciliobrevins during storage has been noted (*Sainath and Gallo, 2015*). The benzoylacrylonitrile core is required for in vitro and cellular activity (*Firestone et al., 2012*; *See et al., 2016*). This functional group has the potential to isomerize, and the ciliobrevin scaffold may exist as either of two isomers about the benzoylacrylonitrile olefin (C2 - C9, *Figure 1A*). The preferred isomer of this compound has not been determined and chemical modification of the quinazolinone or acyl groups, even distal to the acrylonitrile functionality, has the potential to affect the geometry of the compound's core. Together, these factors have made activity-guided modifications to improve compound potency challenging. Further, up to ~100 fold differences in potency have been noted between biochemical and cell based assays (0.2 µM – 30 µM for ciliobrevin D), raising concerns about target specificity. Design and chemical synthesis of alternative scaffolds that address these limitations and retain activity against dynein are needed.

Dynein is a large, ~4600 amino acid protein that contains six unique AAA ATPase domains (*Carter, 2013*). Many AAA+ enzymes function as homohexameric arrays of identical AAA domains, and thus all six of the ATPase sites can be biochemically equivalent (*Erzberger and Berger, 2006*). However, for dyneins, the six unique AAA domains reside on a single polypeptide and each can have a specialized role in motor protein function (*Carter, 2013*). In the case of dynein 1, four of the six AAA domains contain the residues necessary for nucleotide binding; of these, only

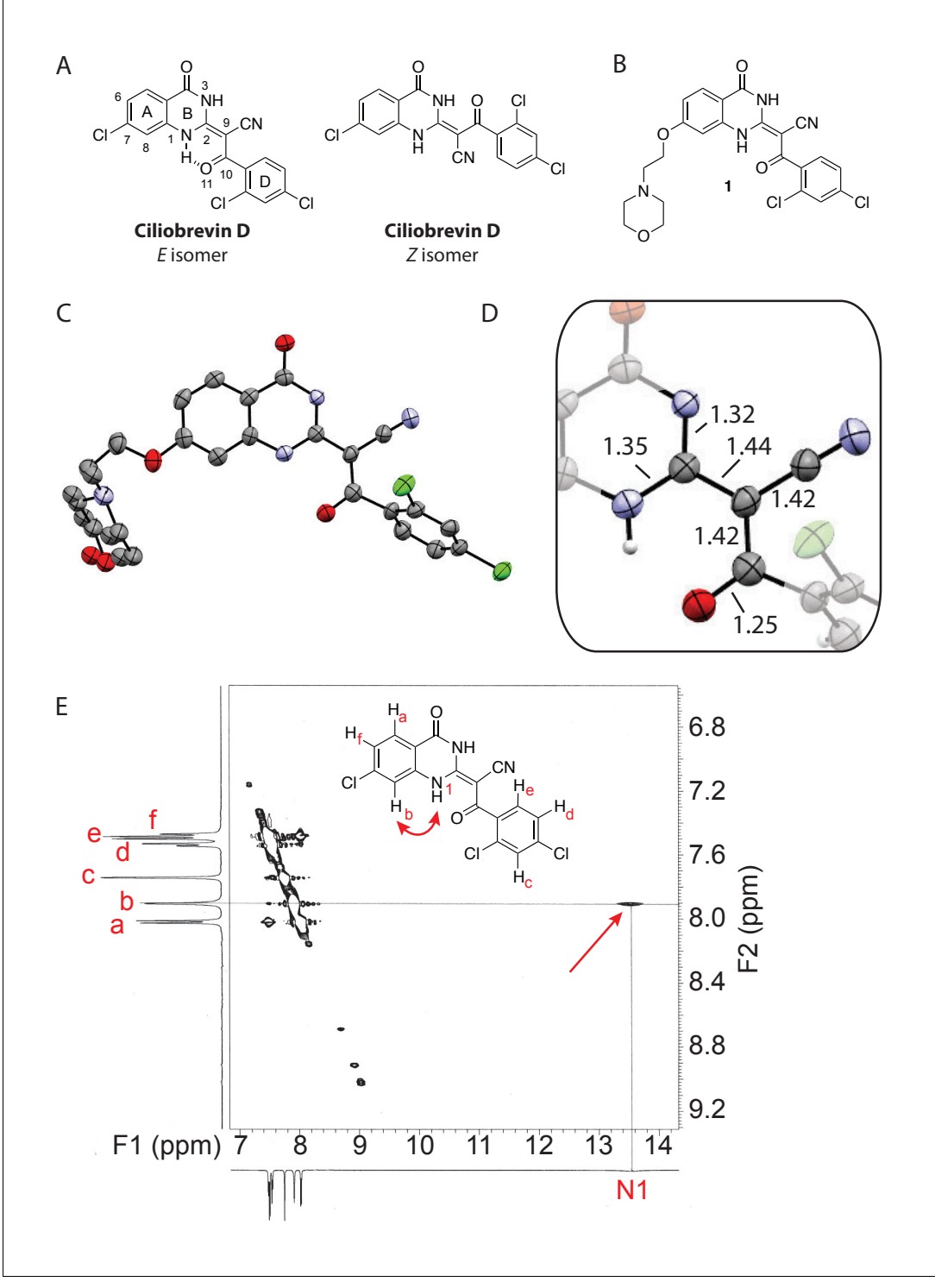

**Figure 1.** Analysis of the conformation of the ciliobrevin scaffold. (**A**) *E* and *Z* isomers about the C2-C9 bond of ciliobrevin D are shown. Possible hydrogen-bond in the *E* configuration is indicated (dashed line). Selected atoms are numbered for reference. (**B**) Compound **1** was used for x-ray crystallography. (**C**) X-ray structure of **1**. Displacement ellipsoids are shown at the 50% probability level. (**D**) Enlarged (2x) image of acrylonitrile moiety with selected bond lengths indicated (Å). Protons are shown to illustrate possible hydrogen-bonding interaction. Color legend: carbon-grey, hydrogen-white, nitrogen-blue, oxygen-red, chlorine-green. (**E**) Nuclear Overhauser effect spectroscopy (NOESY) spectrum for ciliobrevin D. A cross-peak corresponding to interaction between $H_b$ and the N1 proton is indicated with a single-headed arrow. Protons corresponding to peaks in the spectrum of ciliobrevin

*Figure 1 continued on next page*

*Figure 1 continued*

D are indicated. Coupling is indicated by a double-headed arrow. A one-dimensional proton NMR spectrum of ciliobrevin D is shown in *Figure 1—figure supplement 1*.
The following figure supplement is available for figure 1:

**Figure supplement 1.** $^1$H NMR spectrum of ciliobrevin D.

AAA1 and AAA3 substantially contribute to ATP hydrolysis and have been shown to be required for microtubule motility (*Carter, 2013*). Dynein's ATPase activity is stimulated by interactions with microtubules and is thought to occur mainly at AAA1, while hydrolysis at AAA3 plays a regulatory role (*Bhabha et al., 2014*; *DeWitt et al., 2015*; *Nicholas et al., 2015a*). Earlier work suggested that ciliobrevins are ATP-competitive inhibitors of dynein (*Firestone et al., 2012*), but it remains unclear which of the six AAA sites are modulated, adding to the challenges of inhibitor optimization.

Here, we characterize the conformation of ciliobrevins and design tricyclic pyrazoloquinazolinone derivatives that are more potent inhibitors of dynein. One derivative, which we name dynapyrazole-A inhibits dyneins 1 and 2 with similar potencies in vitro and in cellular assays. Biochemical analyses of this compound showed that while it inhibits the microtubule-stimulated ATPase activity of dynein, it does not potently block the microtubule-independent basal activity. This mode of activity is unlike that of the ciliobrevins, which inhibit both basal and microtubule-stimulated hydrolysis.

## Results

To design new analogs with improved properties, we analyzed the conformation of the ciliobrevin scaffold. For these studies, we first used X-ray crystallography. Efforts to crystallize ciliobrevin D were unsuccessful; however, compound **1**, a derivative with a 2-morpholinoethyl ether substitution, was synthesized using a previously described procedure and was found to readily crystallize (*Figure 1B,C,D*) (*See et al., 2016*). The X-ray data suggest that compound **1** exists as a single isomer with an *E*-olefin configuration of the C2-C9 double bond. The measured bond lengths are consistent with electrons in a π-system delocalized across the benzoylacrylonitrile core (*Figure 1D*). The C2-C9 bond length (1.44 Å), which is longer than that of a typical olefin, indicates significant single bond character. The proximity between O11 and N1-H suggests that the ciliobrevin structure is stabilized by an intramolecular hydrogen-bond between N1 and O11 (*Figure 1A and D*). A closely related acylacrylonitrile-substituted quinazolinone was found to favor an alternative conformation, corresponding to a Z-olefin at C2-C9, that was also stabilized by intramolecular hydrogen-bonding (*Milokhov et al., 2013*). All these data together indicate it is likely that the isomeric preference of the pharmacophore is sensitive to distal substitutions.

As crystal packing may impact conformation, we turned to NMR spectroscopy to analyze the structure of the ciliobrevin scaffold in solution. A one-dimensional NMR spectrum of ciliobrevin D showed a peak at 13.5 ppm, which could be assigned to a proton at one of the quinazolinone nitrogens (*Milokhov et al., 2013*) (*Figure 1—figure supplement 1*). The presence of this broad, downfield peak suggests that one exchangeable N-H proton is stabilized by a hydrogen-bonding interaction. A NOESY spectrum of ciliobrevin D revealed all the expected resonances (*Figure 1E*). In addition, we detected a coupling between the proton at 13.5 ppm and the proton at the 8-position of the quinazolinone, consistent with the proton at N1 being involved in hydrogen bonding. Together, these data suggest that ciliobrevin D has similar orientation in solution to that of **1** in the crystal, and the benzoylacrylonitile functional group favors an *E*-isomer configuration that is likely to be stabilized by hydrogen bonding (*Figure 1A and E*).

We hypothesized that replacing the benzoylacrylonitrile core with a heterocyclic scaffold that maintained the observed ciliobrevin geometry could lead to improved dynein inhibitors. We reasoned that a tricyclic scaffold could replace the non-covalent N1-H-O11 interaction and maintain the overall ciliobrevin pharmacophore. We envisioned replacing the C2-C9 olefin and C10 ketone in ciliobrevin with either a pyrrole or pyrazole ring (*Figure 2A–C*). We adapted established procedures to synthesize **2** and **3**, which differ from known compounds only in the substitution pattern of the D ring (*Figure 2A and B*) (*Süsse and Johne, 1981*; *Orvieto et al., 2009*). We also

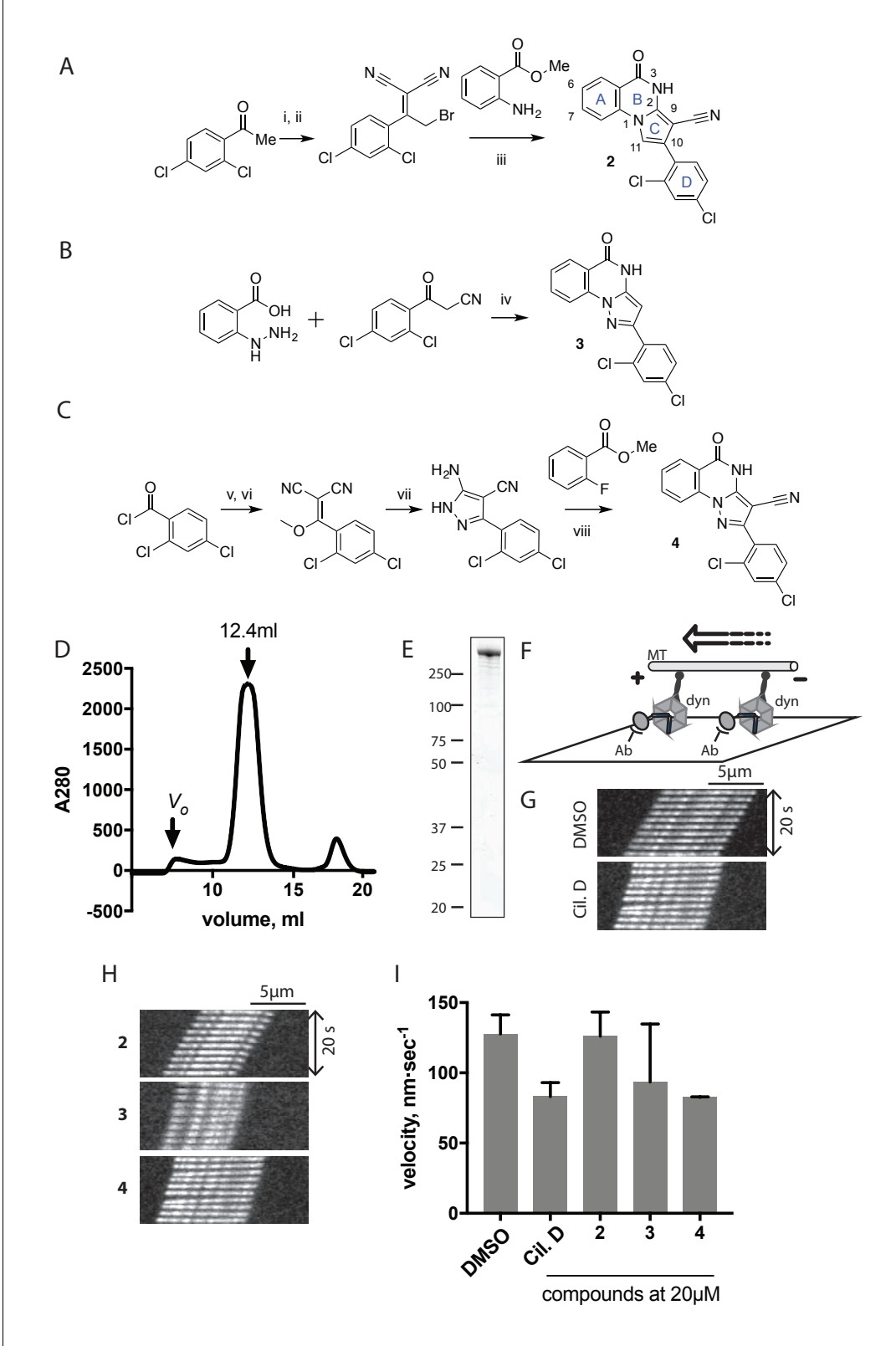

**Figure 2.** Synthesis of ciliobrevin D derivatives and analysis of their activity against dynein 2. (A–C) Synthesis of ciliobrevin derivatives. (A) Pyrroloquinazolinone derivative. Reagents, conditions, and yield (%): (i) malononitrile, ammonium acetate, toluene, 100°C, 13 hr, 72%; (ii) bromine, carbon tetrachloride, 70°C, 9 hr, 38%; (iii) methyl anthranilate, isopropanol, 100°C, 20 hr, 15%. Selected atoms are numbered for reference. (B–C)
*Figure 2 continued on next page*

*Figure 2 continued*

Pyrazoloquinazolinone derivatives. Reagents, conditions, and yield (%): (iv) acetic acid, 150°C (microwave), 30 min, 26%. (v) malononitrile, sodium hydride, tetrahydrofuran, 0°C, 1 hr, 96%; (vi) dimethyl sulfate, *N,N*-diisopropylethylamine, dioxane, 60°C, 23 hr, 27%; (vii) hydrazine hydrate, ethanol, 80°C, 6 hr, 82%; (viii) methyl 2-fluorobenzoate, potassium carbonate, dimethylformamide, 140°C, 30 min, 11%. (D) Gel filtration trace (Superose 6) for GFP-dynein 2, with volume at elution peak indicated. $V_o$, void volume. (E) SDS-PAGE analysis (Coomassie blue stain) of GFP-dynein 2, ~0.5 µg protein loaded. (F) Schematic of microtubule motility assay. Anti-GFP antibody (Ab), GFP-dynein (dyn), and microtubule (MT) are indicated. (G) Montages of fluorescent microtubules moving on GFP-dynein-2-coated glass slides in the solvent control (2% DMSO) or in the presence of ciliobrevin D (20 µM). (H) Montages of fluorescent microtubules moving on GFP-dynein-2-coated glass coverslips in the presence of compounds **2–4** (20 µM). (I) Mean velocity of dynein-2-driven microtubule gliding in the presence of control solvent (2% DMSO), ciliobrevin D, or compounds **2–4** (mean. + S.D., n $\geq$ 3). Number of microtubules quantified: DMSO-327, Cil. D-85, **2**–98, **3**–90, **4**–77. All motility assays were run at 1 mM MgATP, 0.05 mg/mL casein, and 2% DMSO. For all montages, the interval between successive images is 2 s and total time elapsed is 20s. Horizontal scale bar, 5 µm.

devised a synthetic route to a series of tricyclic analogs exemplified by **4**. This synthesis relies on the condensation of a 2-fluorobenzoic acid methyl ester with a 4-cyano-aminopyrazole under basic conditions (*Figure 2C*). This strategy allowed convergent synthesis of the desired cyclized ciliobrevin analogs.

To test if these compounds inhibit dynein, we employed a microtubule gliding assay. We first focused on human cytoplasmic dynein 2, the isoform involved in ciliary transport and Hedgehog signaling (*Goetz and Anderson, 2010*), as its inhibition in this assay by ciliobrevin D has not been previously demonstrated. We purified an N-terminally GFP-tagged motor-domain construct of dynein 2 using an insect cell expression system (*Schmidt et al., 2015*). The GFP tag in this construct (GFP-dynein 2, amino acids 1091–4307) allowed the motor protein to be immobilized on passivated glass coverslips used in microtubule gliding assays. GFP-dynein 2 was obtained as a mono-disperse peak by gel filtration (*Figure 2D and E*). In the presence of ATP (1 mM), this protein moved microtubules with a velocity of 128 ± 14 nm/s (mean ± SD, *Figure 2G and I*). This activity is readily revealed by a time-series montage, which shows a fluorescently labeled microtubule being displaced ~3 µm in 20 s (*Figure 2G*). This rate is comparable to previous analyses of this construct (*Schmidt et al., 2015*). Ciliobrevin D (20 µM) reduced the microtubule gliding velocity to 83 ± 10 nm/s (*Figure 2G and I*), consistent with its inhibition of dynein-2-dependent processes in cells (*Firestone et al., 2012*; *Ye et al., 2013*).

We next tested compounds **2–4** at 20 µM. Compound **2** did not substantially change microtubule gliding velocity (126 ± 17 nm/s, *Figure 2H and I*). Compounds **3** and **4** inhibited dynein-2-driven gliding to a velocity comparable to that observed in the presence of ciliobrevin D at this same concentration.

To improve the potency of compound **4**, we compared its structure with that of ciliobrevin derivative compound **1**. In particular, we superimposed the crystal structure of **1** with an energy-minimized structure of **4** computationally generated using Maestro (Schrödinger). This analysis revealed that the dichlorophenyl D ring of **4** projects from the pyrazoloquinazolinone core such that it is offset from the D ring of ciliobrevin (*Figure 3A*). We reasoned that adding a single carbon atom between the pyrazoloquinazolinone and the phenyl ring may lead to closer alignment with ciliobrevin and that using a cyclopropyl spacer would restrict rotation of the resulting scaffold. Using our modular synthesis strategy for pyrazoloquinazolinones, we generated a set of ciliobrevin derivatives with a cyclopropyl group separating the aromatic ring systems (**5–8**, *Figure 3B*). Three of these compounds had substitutions at the 6-position of the quinazolinone (**6**, **7**, and **8**, *Figure 3B*), a modification that we have previously reported to improve the potencies of ciliobrevins (*Firestone et al., 2012*). The crystal structure of **5** indicates that its D ring aligns with that of compound **1** better than the chlorophenyl ring of compound **4** (*Figure 3A, 3C, and 3D*).

We next tested whether compounds **5–8** (20µM) inhibited dynein-2-dependent microtubule gliding (*Figure 3E and F*). Compound **5** reduced velocity to 58 ± 14 nm/s, while addition of a trifluoromethyl group at the 6-position of the 'A' ring in compound **6** led to near-complete inhibition of gliding (6 ± 2 nm/s). By contrast, compound **7**, with a methoxy group at the 6-position of the A ring,

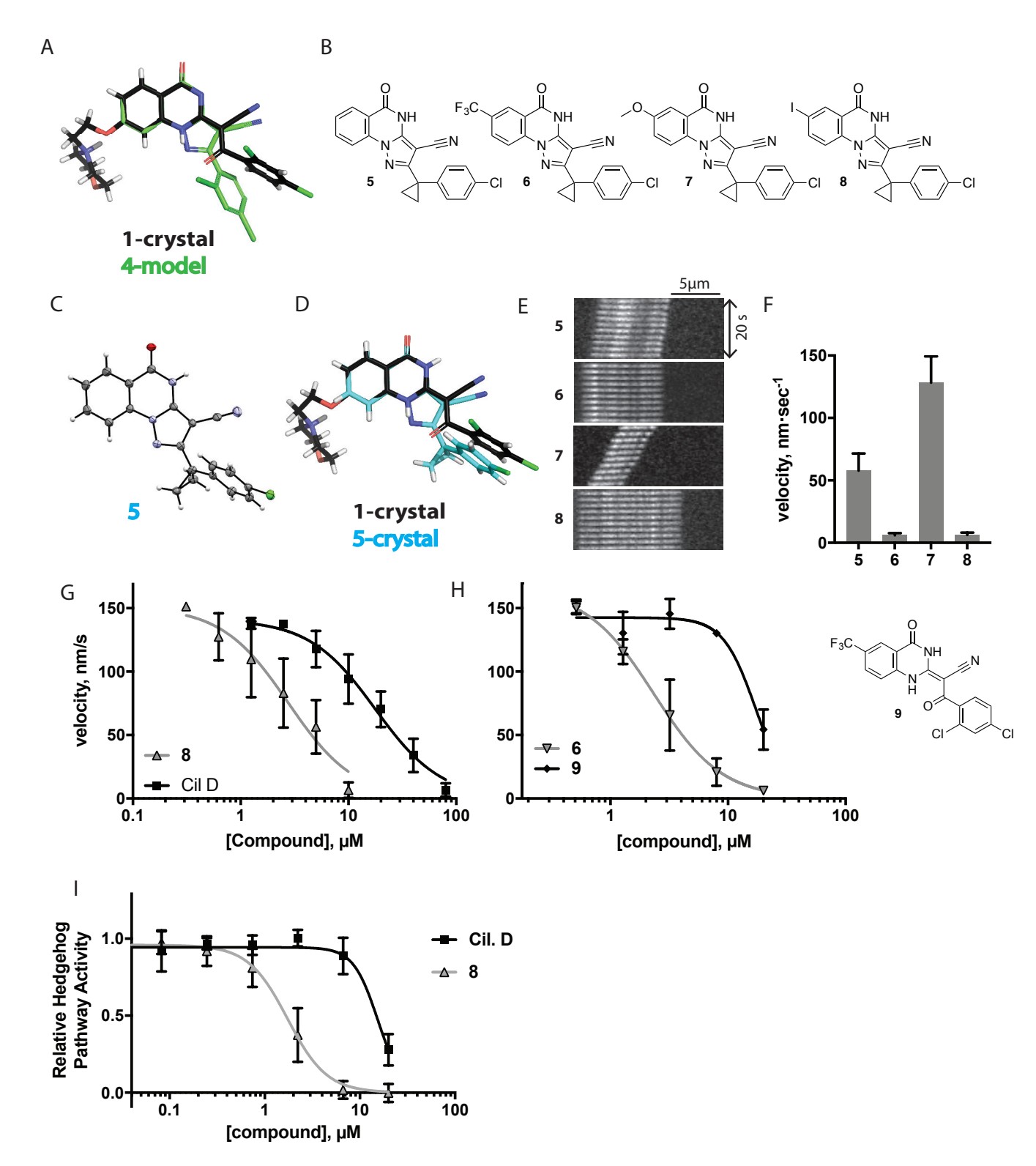

**Figure 3.** Chemical structure analysis, design, and evaluation of pyrazoloquinazolinone derivatives of ciliobrevin. (A) Superposition of the crystal structure of **1** with the computational model of **4** (green). (B) Pyrazoloquinazolinone ciliobrevin derivatives with cyclopropyl group. (C) Crystal structure of **5**. Displacement ellipsoids are shown at the 50% probability level. (D) Superposition of the crystal structure of **1** with the crystal structure of **5** (cyan). (E) Montages of fluorescent microtubules moving on GFP-dynein-2-coated glass coverslips in the presence of compounds **5–8** (20 µM). The interval

*Figure 3 continued on next page*

*Figure 3 continued*

between successive images is 2 s and total time elapsed is 20s. Horizontal scale bar, 5 µm. (F) Velocity of dynein-2-driven microtubule gliding in the presence of compounds 5–8 (mean + S.D., n ≥ 3). Number of microtubules quantified: 5–91, 6–63, 7–86, 8–56. (G) Inhibition of GFP-dynein-2-driven motility by 8 and ciliobrevin D. $IC_{50}$ values: 8: 2.6 ± 1.3 µM (mean ± S.D., n = 3); ciliobrevin D: 20 µM (range: 19–21 µM, n = 2). Velocity distribution histograms for inhibition of dynein-2-driven microtubule motility are presented in *Figure 3—figure supplement 1*. Number of microtubules quantified: 8: 10 µM-36, 5 µM-59, 2.5 µM-98, 1.3 µM-112, 0.6 µM-102, 0.3 µM-126; Ciliobrevin D: 80 µM-10, 40 µM-47, 20 µM-78, 10 µM-85, 5 µM-99, 2.5 µM-66, 1.3 µM-80; (H) Inhibition of GFP-dynein-2-driven motility by 6 and 9. The chemical structure of 9 is shown. $IC_{50}$ values: 6: 2.9 ± 0.6 µM (mean ± S.D., n = 3). 9: 17.7 µM (range: 17.2–18.2, n = 2). Number of microtubules quantified: 6: 20 µM-38, 8 µM-24, 3.2 µM-48, 1.3 µM-50, 0.5 µM-53; 9: 20 µM-29, 8 µM-54, 3.2 µM-50, 1.3 µM-54, 0.5 µM-56. (I) Dose-dependent inhibition of Gli-driven luciferase reporter expression by ciliobrevin D and compound 8. $IC_{50}$ values (mean ± S.D.): ciliobrevin D: 15.5 ± 3 µM (n = 4); 8: 1.9 ± 0.6 µM (n = 5). For G, H, and I, $IC_{50}$ values reported reflect the mean (with range if n = 2 or S.D. if n ≥ 3) of separate $IC_{50}$ values obtained from independent dose-response analyses. Data were fit to a sigmoidal dose-response curve and the fit was constrained such that the value at saturating compound = 0. Individual data points presented reflect mean of values determined from n ≥ 2 independent replicates ± S.D. (G, I) or ± range (H). All motility assays were performed at 1 mM MgATP, 0.05 mg/mL casein, and 2% DMSO.

The following figure supplement is available for figure 3:

**Figure supplement 1.** Microtubule velocity distribution histograms for dynein-2-driven microtubule gliding in the presence of different concentrations of compound 8.

did not substantially inhibit dynein-2-driven microtubule gliding (128 ± 21 nm/s, *Figure 3E and F*). Compound 8, with a 6-iodo substituent at the A ring, showed comparable activity to 6 (6 ± 2 nm/s). For the two most active compounds (6 and 8) as well as ciliobrevin D, we performed dose-dependent analyses. We found six- to eightfold increases in potencies of compounds 6 ($IC_{50}$: 2.9 ± 0.6 µM) and 8 ($IC_{50}$: 2.6 ± 1.3 µM) relative to ciliobrevin D ($IC_{50}$ of 20 ± 1.0 µM) in this assay (*Figure 3G,H*). To our knowledge, compound 8 is the most potent inhibitor of dynein 2 published to date. Hereafter, we designate compound 8 as dynapyrazole-A and compound 6 as dynapyrazole-B.

In order to directly test the effect of replacing the benzoylacrylonitrile-quinazolinone system by a pyrazoloquinazolinone, we also synthesized uncyclized congeners of dynapyrazoles. Due to poor solubility of a 6-iodo-substituted ciliobrevin derivative it was not possible to test its activity and compare it with that of compound 8. Comparisons between compound 6 (dynapyrazole-B) and compound 9, a 6-CF3-substituted ciliobrevin derivative (*Figure 3H*), show that cyclization leads to ~6-fold improvement in the potency of dynein 2 inhibition in the microtubule gliding assay.

We next examined the inhibition of dynein 2 by dynapyrazole-A in cell-based assays. In cell culture, serum starvation results in the formation of primary cilia, which are required for Hedgehog signaling (*Goetz and Anderson, 2010*). Quiescent cells respond to the Hedgehog ligand or to a synthetic agonist and pathway activity can be measured using a Gli-driven luciferase reporter (*Taipale et al., 2000*; *Chen et al., 2002*). Expression of Hedgehog-driven luciferase reporter was inhibited by dynapyrazole-A (Compound 8, $IC_{50}$: 1.9 ± 0.6 µM) ~8 fold more potently than ciliobrevin D ($IC_{50}$: 15.5 ± 3 µM, *Figure 3I*). Cell death was observed at concentrations 10-fold above the $IC_{50}$ for dynapyrazole-A (20 µM) over the 28 hr time course of this experiment. In this assay, we have also observed cell death at high ciliobrevin D concentrations (200 µM,~10x above its $IC_{50}$). To stimulate Hedgehog pathway activity, we used the synthetic agonist of smoothened (SAG) (*Chen et al., 2002*), which competes with many known Hedgehog pathway inhibitors for binding to Smoothened, a key protein in this signaling pathway (*Sharpe et al., 2015*). As dynapyrazole-A inhibits Hedgehog signaling at high SAG concentrations (500 nM), it is likely to act downstream of Smoothened, as was previously noted for ciliobrevins and consistent with inhibition of dynein 2 (*Hyman et al., 2009*; *Firestone et al., 2012*).

The Hedgehog signaling pathway depends on dynein-2-driven retrograde intraflagellar transport in the primary cilium (*Goetz and Anderson, 2010*). We used time-lapse microscopy to examine the dynamics of fluorescently labeled intraflagellar transport protein-88 (mNeonGreen-IFT88, *Figure 4A–C*, *Videos 1–4*), an approach we have previously used to study the effect of ciliobrevins and derivatives (*Ye et al., 2013*; *See et al., 2016*). Time-lapse recordings of cells treated with control media revealed a steady flow of IFT88 punctae moving from cilium tip to base and vice versa, as expected (*Videos 1* and *2*). The wider end of the cilium was identified as its base, a convention suggested previously by others (*Yang et al., 2015*). We find that in cilia treated with dynapyrazole-A

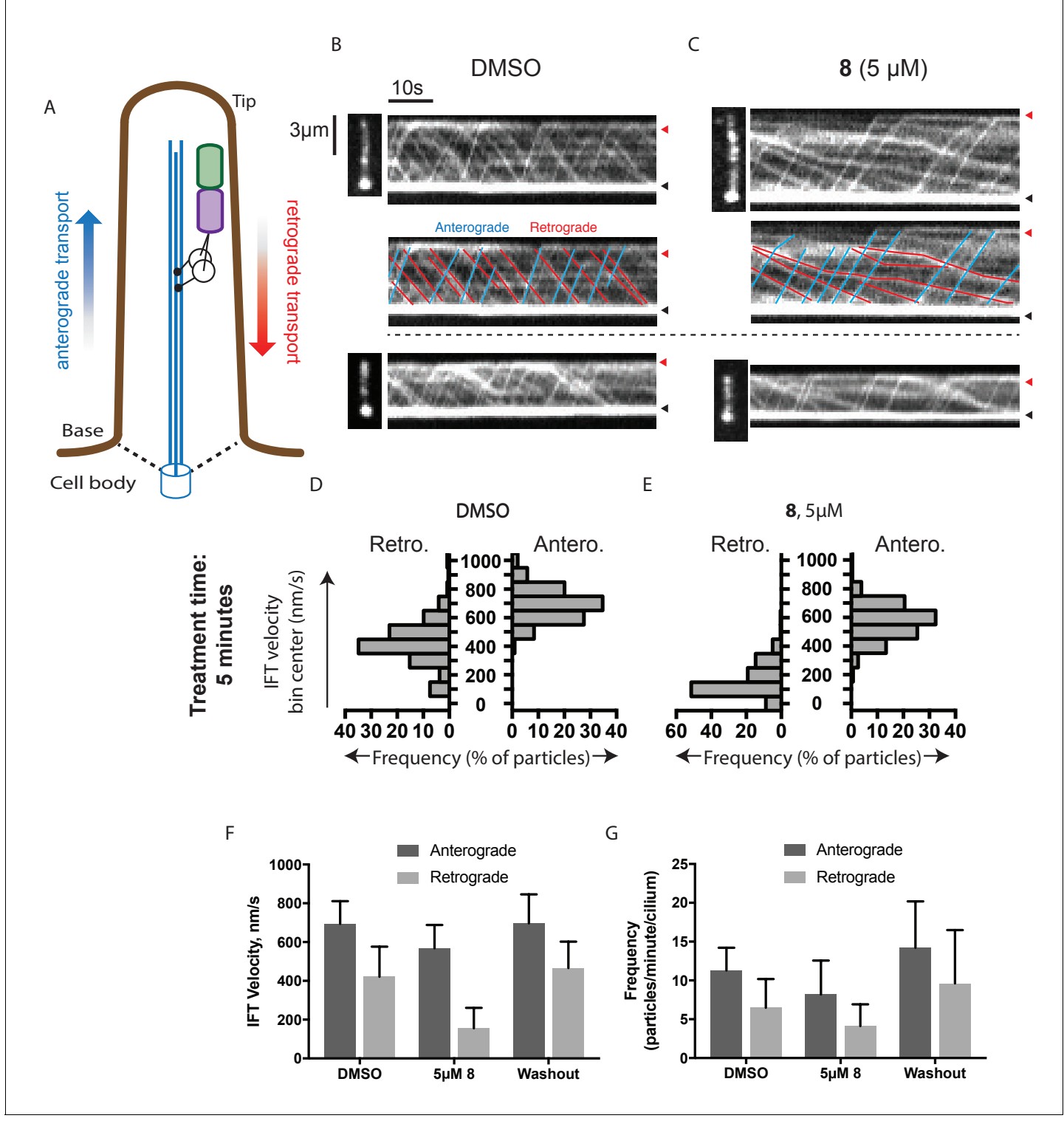

**Figure 4.** Analysis of the effect of dynapyrazole-A (**8**) on intraflagellar transport. (**A**) Schematic of cilium showing microtubule-based axoneme (blue) and dynein (black), and an intraflagellar transport particle (purple) containing mNeonGreen-IFT88 (green). Anterograde and retrograde transport directions are indicated. (**B–C**) Images from time-lapse series and associated kymographs showing motion of mNeonGreen-IFT88-containing particles in primary cilia of living murine kidney cells (IMCD3). Cilium tip (red arrowhead) and base (black arrowhead) are indicated. Red (retrograde) and blue (anterograde) traces have been added to one kymograph per condition to illustrate particle tracks. Image scale bar, 3 µm; Kymograph horizontal scale bar, 10 s; kymograph vertical scale bar, 3 µm. (**D–E**) Velocity distribution histograms showing anterograde and retrograde velocities in the solvent control (0.3% DMSO, **D**) and in the presence of 5 µM **8** (**E**) at 5 min after initiation of experiment. Analyses of cells treated with 5 µM **8** for 10 min are shown in

*Figure 4 continued on next page*

*Figure 4 continued*

*Figure 4—figure supplement 1*. Analyses of cells treated with 10 µM **8** are shown in *Figure 4—figure supplement 2*. (F–G). Intraflagellar transport velocities (F) and frequencies (G) after washout of dynapyrazole-A. Bars represent mean + S.D. Data analysis ($V_a$, anterograde velocity, nm/s; $V_r$, retrograde velocity, nm/s; $F_a$, anterograde frequency, counts/minute; $F_r$, retrograde frequency, counts/minute; values are mean ± S.D. $N_a$, number of anterograde particles analyzed; $N_r$, number of retrograde particles analyzed; C, number of cilia analyzed). DMSO, $V_a$ 694 ± 117, $V_r$ 421 ± 156, $F_a$ 11.3 ± 3, $F_r$ 6.5 ± 4, $N_a$ 429, $N_r$ 244, C 38; 5 µM **8**, $V_a$ 566 ± 116, $V_r$ 156 ± 107, $F_a$ 8.5 ± 5, $F_r$ 4.1 ± 3, $N_a$ 443, $N_r$ 211, C 52; Washout, $V_a$ 697 ± 149, $V_r$ 467 ± 136, $F_a$ 14.2 ± 6, $F_r$ 9.6 ± 7, $N_a$ 256, $N_r$ 173, C 18.

The following figure supplements are available for figure 4:

**Figure supplement 1.** Analysis of intraflagellar transport following 10 min exposure to 5 µ M dynapyrazole-A (compound **8**).

**Figure supplement 2.** Analysis of intraflagellar transport at 10 µM dynapyrazole-A (compound **8**).

**Figure supplement 3.** The effect of serum concentration on washout of dynapyrazole-A (compound **8**) in intraflagellar transport assays.

(compound **8**, 5 µM), retrograde-directed IFT88 punctae were markedly slowed (*Videos 3* and *4*). In contrast, anterograde motion did not appear to be substantially altered in the presence of dynapyrazole-A.

To quantitatively assess the effect of dynapyrazole-A treatment on intraflagellar transport, we analyzed time-lapse images of cilia using KymographDirect, an automated analysis algorithm that extracts particle velocities from kymographs (*Mangeol et al., 2016*). Previous analyses of the effect of dynein 2 inhibition using a temperature-sensitive mutant in *Chlamydomonas* revealed that dynein 2 depletion causes a ~60–70% reduction in retrograde velocities and a ~20% reduction in anterograde velocities as well as 30–60% reductions in the frequency of particle transport in both directions (*Engel et al., 2012*). Under control conditions (0.3% DMSO, *Figure 4B*), anterograde particles moved with a speed of 694 ± 117 nm/s (*Figure 4D and F*, mean ± S.D., 429 particles, 38 cilia) and retrograde particles moved at 421 ± 156 nm/s (*Figure 4D and F*, 244 particles, 38 cilia), consistent with previous studies (*Ye et al., 2013*). Following addition of dynapyrazole-A to cells, the speed of retrograde particles was markedly reduced at five minutes, the fastest reliable time line for this experiment on our microscopy set-up (*Figure 4C,E and F*; 5 µM compound **8**: mean velocity 156 ± 107 nm/s, 211 particles, 52 cilia). In contrast, anterograde particle velocities were only reduced by ~18% (*Figure 4C,E and F*, 5 µM **8**: 566 ± 116 nm/s, 443 particles, 52 cilia). After 10 min of treatment, reductions in velocities were similar to those at the 5 min time point (*Figure 4—figure supplement 1*). Treatment of cilia with a higher dynapyrazole-A concentration (10 µM) slowed both retrograde- and anterograde-directed motion (*Figure 4—figure supplement 2*). Again, retrograde motion was more strongly inhibited. Dynapyrazole-A treatment (5 µM and 10 µM) also reduced the frequency, that is, the number of particles moving across a cilium per minute, in both anterograde and retrograde directions (*Figure 4G*, *Figure 4—figure supplement 2*). We note that dynapyrazole-A, at concentrations close to the $IC_{50}$ for inhibiting microtubule gliding in vitro, alters intraflagellar transport in a manner similar to what has been observed following dynein 2 loss-of-function in *Chlamydomonas* (*Engel et al., 2012*).

We next examined whether inhibition of intraflagellar transport by dynapyrazole-A was reversed following washout of the compound. Ciliated cells treated with dynapyrazole-A (5 µM compound **8**, 5 min) were transferred to solvent-control media with serum (0.3% DMSO, 10% FBS) and incubated for an additional 10 min. Both retrograde and anterograde velocities recovered to control levels (*Figure 4F*, velocities following washout: retrograde: 467 ± 136 nm/s, 173 particles, 18 cilia; anterograde: 697 ± 149 nm/s, 256 particles, 18 cilia) as did transport frequencies (*Figure 4G*). When media with a lower

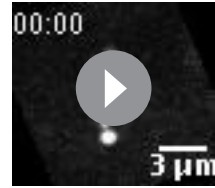

**Video 1.** Dynamics of intraflagellar transport particles. Timelapse movie of mNeonGreen-IFT88 particles in IMCD3 cells under solvent control conditions (0.3% DMSO). Scale bar, 3 µm. Time, min:sec.

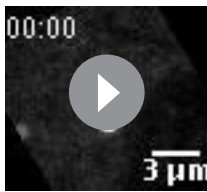

**Video 2.** Dynamics of intraflagellar transport particles. Timelapse movie of mNeonGreen-IFT88 particles in IMCD3 cells under solvent control conditions (0.3% DMSO). Scale bar, 3 μm. Time, min:sec.

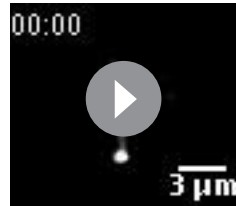

**Video 3.** Dynamics of intraflagellar transport particles. Timelapse movie of mNeonGreen-IFT88 particles in IMCD3 cells in the presence of compound **8** (5 μM). Scale bar, 3 μm. Scale bar, 3 μm. Time, min:sec.

serum concentration was used in washout experiments, retrograde velocities recovered only partially, suggesting that serum may accelerate the partitioning of this compound out of cells (*Figure 4—figure supplement 3*). Taken together, our data suggest dynapyrazole-A is likely to be a useful reversible probe to study intraflagellar transport.

We predicted that dynapyrazole-A, like ciliobrevin D, should also inhibit cytoplasmic dynein 1 (*See et al., 2016*). To examine the inhibition of dynein 1 by dynapyrazole-A in vitro we generated recombinant human protein. We expressed and purified a GFP-tagged human dynein 1 (AA 1320–4646) construct similar to the one we used for GFP-dynein 2. This protein migrated with a peak elution volume of 12.6 mL in size exclusion chromatography and SDS-PAGE analysis showed >90% purity (*Figure 5A,B*). GFP-dynein 1 moves microtubules at 508 ± 60 nm/s (n = 5 independent experiments, 191 microtubules analyzed), a velocity expected based on studies of other mammalian dynein 1 homologs (*Yamada et al., 2008*). Under these conditions, ~97% of the filaments analyzed had velocities >50 nm/s (*Figure 5—figure supplement 1*). Montages showed that both ciliobrevin D and dynapyrazole-A slowed dynein-dependent microtubule gliding (*Figure 5C*). Dose-dependent analysis indicated that dynapyrazole-A blocked GFP-dynein-1-driven motility with an $IC_{50}$ of 2.3 ± 1.4 μM,~6-fold more potently than ciliobrevin D (15 ± 2.9 μM, *Figure 5D*). Inhibition of dynein-1-dependent microtubule gliding by dynapyrazole-A was reversed following washout, as is also the case for dynein-2-dependent motility (*Figure 5—figure supplement 3*). As has been previously noted for ciliobrevin D, the potency of dynapyrazole-A was sensitive to the protein (e.g. blocking agent, serum) concentration in solution (*Figure 5—figure supplement 4*), likely due to the hydrophobicity of these compounds (calculated logarithm of octanol:water partition coefficient [ClogP]: ciliobrevin D: 4.5; dynapyrazole-A: 4.2) (*Firestone et al., 2012*). We therefore focused our cell-based analyses of dynapyrazole-A on dynein-1-dependent processes in assays that do not require high serum concentrations.

Lysosome transport can be observed in the neurites of CAD cells, a murine cell line that displays neuron-like properties in serum-free cell culture media (*Figure 5E–M* and *Videos 5–7*) (*Qi et al., 1997*). This bidirectional transport of lysosomes requires kinesins and dynein 1 and can be observed by imaging live cells stained with the LysoTracker dye (*Vale, 2003*; *Pu et al., 2016*). A coupling between anterograde and retrograde motion has been described for membrane-bound organelles,

with disruption of dynein-driven motion blocking bi-directional organelle motility (*Barlan et al., 2013*). We imaged control solvent (0.1% DMSO) treated cells and observed bidirectional motion of LysoTracker-labelled puncta (*Figure 5E,H,K*, *Video 5*). Overlays of successive images from a time-lapse series, color-coded for displacement, reveal organelle displacements (*Figure 5H–J*). Quantitative analysis using kymographs shows that ~20% of particles move at speeds below 25 nm/s in both anterograde and retrograde directions under control conditions and a similar fraction of particles have velocities above 250 nm/s

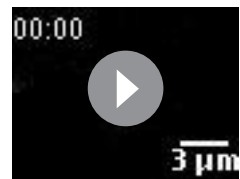

**Video 4.** Dynamics of intraflagellar transport particles. Timelapse movie of mNeonGreen-IFT88 particles in IMCD3 cells in the presence of compound **8** (5 μM). Scale bar, 3 μm. Scale bar, 3 μm. Time, min:sec.

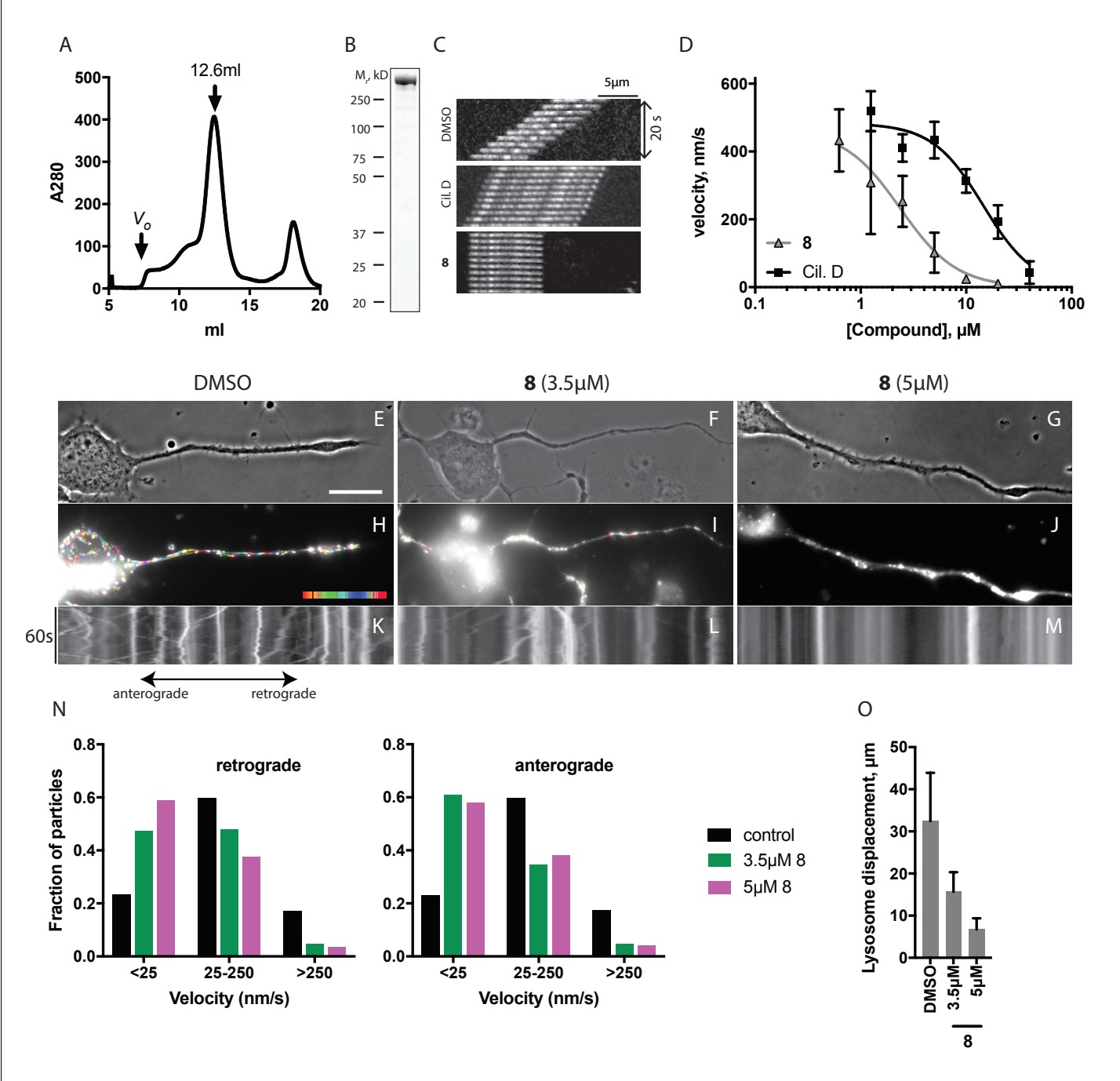

**Figure 5.** Inhibition of dynein 1 activity by dynapyrazole-A (compound **8**). (**A**) Gel filtration trace (Superose 6) for GFP-dynein 1, with volume at elution peak indicated. $V_o$, void volume. (**B**) SDS-PAGE analysis (Coomassie blue stain) of GFP-dynein 1, ~0.5 µg protein loaded. (**C**) Montages of fluorescent microtubules moving on GFP-dynein 1-coated glass coverslips in the presence of 1 mM ATP and either DMSO, ciliobrevin D or **8** (10µM). The interval between successive images is 2 s and total time elapsed is 20s. Horizontal scale bar, 5 µm. (**D**) Inhibition of GFP-dynein 1-driven motility by **8** and ciliobrevin D (mean ± S.D., n = 3). $IC_{50}$ values for **8**: 2.3 ± 1.4 µM (n = 3); ciliobrevin D: 15 ± 2.9 µM (n = 3). Number of microtubules quantified: **8**: 20 µM-98, 10 µM-105, 5 µM-108, 2.5 µM-97, 1.3 µM-134, 0.6 µM-99, 0.3 µM-43, 0.2 µM-29; ciliobrevin D: 40 µM-64, 20 µM-74, 10 µM-82, 5 µM-79, 2.5 µM-81, 1.3 µM-87. $IC_{50}$ values reported reflect the mean (± S.D.) of separate $IC_{50}$ values obtained from independent dose-response analyses. Data were fit to a sigmoidal dose-response curve and the fit constrained such that the value at saturating compound >0. All motility assays were performed at 1 mM MgATP, 0.05 mg/mL casein, and 2% DMSO. Velocity distribution histograms for inhibition of dynein-1 driven microtubule motility are presented in *Figure 5—figure supplement 1*. Analysis of microtubule attachment to dynein-coated coverslips is presented in *Figure 5—figure supplement 2*. (E - J) Images of CAD cell neurites stained with Lysotracker Red. in the presence of DMSO control (0.1%), 3.5 µM and 5 µM (**8**). Scale bar, 10 µm. (**E–G**)

*Figure 5 continued on next page*

*Figure 5 continued*

Phase contrast microscopy images of CAD cells. (**H–J**) Overlay of successive images of lysosome motility in CAD cell neurites. Sixty images, spaced 1s apart, are stacked and successive images colored using FIJI according to the temporal color code shown. (**K–M**) Kymographs corresponding to images in H-J. The kymograph size is 60 s (vertical) by 37 µm (horizontal) and the anterograde and retrograde orientations are indicated. (**N**) Quantitation of lysosome velocity. (**O**) Quantitation of total lysosome displacement over the time course of imaging (1 min). Data are mean of n $\geq$ 2 experiments with $\geq$150 particles counted per experiment. Number of frame-to-frame velocities measured: DMSO-anterograde: 14167, DMSO-retrograde: 14973, 3.5 µM **8**-anterograde: 11283, 3.5 µM **8**-retrograde: 11340, 5 µM **8**-anterograde: 9449, 5 µM **8**-retrograde: 10458. For O, number of particles counted: DMSO-3770, 5 µM **8**–2400, 3.5 µM **8**–840. Error bars: S.D. (DMSO, 5 µM **8**), or range of values (3.5 µM **8**).

The following figure supplements are available for figure 5:

**Figure supplement 1.** Microtubule velocity distribution histograms for dynein-1-driven microtubule gliding in the presence of different concentrations of compound dynapyrazole-A (compound **8**).

**Figure supplement 2.** Analysis of the number of microtubules associated with coverslips in gliding assays.

**Figure supplement 3.** Reversibility of inhibition by dynapyrazole-A (compound **8**).

**Figure supplement 4.** Analysis of the effect of blocking agent on dynein inhibition by dynapyrazole-A (compound **8**).

**Figure supplement 5.** Analysis of the effect of dynapyrazole-A (compound **8**) on intracellular ATP concentrations.

(*Figure 5K and N*). Treatment with dynapyrazole-A suppressed bidirectional motion. In particular, the percentage of retrograde-directed lysosomes moving slower than 25 nm/s increased to ~47% (3.5µM) and ~59% (5µM) while the proportion of lysosomes moving at speeds > 250 nm/s decreased to below 10% (*Figure 5N*). We observed equivalent effects of dynapyrazole-A on anterograde motion velocities (kymographs in *Figure 5K,L and M*). In control cells, lysosomes with measureable displacements had trajectories that averaged 32 ± 11 µm (*Figure 5O*). Dynapyrazole-A (3.5 µM, *Figure 5I and L*) reduced the average track length of moving particles to 15.7 ± 4.7 µm, which can be seen as white punctae that result from the overlay of multiple color-coded frames without particle translocation. A higher dose of dynapyrazole-A 5 µM) shortened the average track length of moving particles to 6.7 ± 2.7 µm (*Figure 5J,M,O*). We found that ATP levels in cells treated with dynapyrazole-A at concentrations that inhibited lysosome transport (5 µM) were stable for up to 3 hr (*Figure 5—figure supplement 5*). Higher concentrations of dynapyrazole-A (15 µM) lowered intracellular ATP levels by ~70% at 3 hr (*Figure 5—figure supplement 5*). Together, these data suggest that at concentrations near its in vitro IC$_{50}$, dynapyrazole-A suppresses lysosome transport in cells via inhibition of dynein 1.

Dynein 1, whose biochemical activity has been extensively characterized, serves as a model for the mechano-chemistry of this motor protein family. Therefore, to dissect the mechanism of inhibition of dynein by dynapyrazole-A, we focused on this isoform. The basal ATPase activity of GFP-dynein was 0.62 ± 0.2 s$^{-1}$ (*Figure 6A*, n = 8 , mean ± S.D.). We also purified an N-terminally polyhistidine (His)-tagged dynein motor domain construct (AA 1320–4646) that was routinely obtained in five-fold higher yield than the GFP-tagged construct (*Figure 6B*). Following a three-step purification protocol, we obtained this protein with >80% purity. The specific activity of His-dynein 1 (0.66 ± 0.1 s$^{-1}$, n = 11, mean ± S.D. *Figure 6C*) was similar to that of GFP-dynein. The basal ATPase rates of these human dynein constructs were within the range observed by others for mammalian dyneins (*Shpetner et al., 1988*; *Ori-McKenney et al., 2010*; *Nicholas et al., 2015b*). We examined the inhibition of dynein 1's ATPase activity by dynapyrazole-A. Remarkably, a high concentration of dynapyrazole-A (40 µM) did not inhibit the basal ATPase activity of GFP-dynein 1

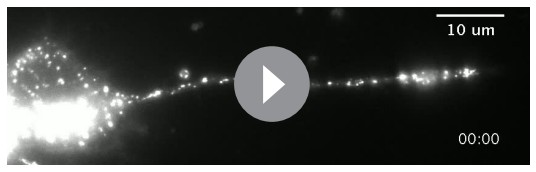

**Video 5.** Lysosome dynamics in CAD cell neurites. Time-lapse movies of Lysotracker Red-treated CAD cells in the solvent control (0.1% DMSO). Time, min:sec.

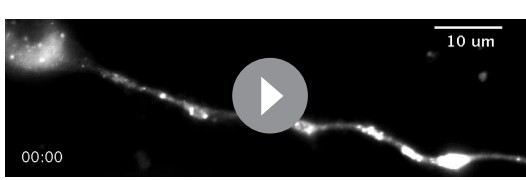

**Video 6.** Lysosome dynamics in CAD cell neurites treated with dynapyrazole-A (compound **8**). Time-lapse movies of Lysotracker Red-treated CAD cells in the presence of dynapyrazole-A (compound **8**, 3.5 µM). Time, min:sec.

(*Figure 6A*) and only partially inhibited His-dynein 1 (~30% inhibition *Figure 6C*). Under similar conditions, ciliobrevin D (40 µM) inhibited the ATPase activity of both GFP-dynein 1 and His-dynein 1 by ~70% (*Figure 6A and C*). The rather modest inhibition of dynein's ATPase activity by dynapyrazole-A was unexpected, as dynapyrazole-A can inhibit microtubule gliding at a 20-fold lower concentration.

We next examined if dynapyrazole-A inhibits dynein 1's microtubule-stimulated ATPase activity. We find that human dynein's ATP hydrolysis rate is stimulated in a microtubule concentration-dependent manner (*Figure 6D*, 2.5 µM: $1.2 \pm 0.05 \text{ s}^{-1}$, n = 3; 8 µM: $1.5 \pm 0.3 \text{ s}^{-1}$, n = 4; 15 µM: $2.2 \pm 0.5 \text{ s}^{-1}$, n = 4). Dynapyrazole-A (30 µM) inhibited the microtubule-stimulated ATPase activity to a residual rate of $\sim 0.5 \text{ s}^{-1}$ (8 or 15 µM microtubules, *Figure 6D*). Dose-dependent analysis was carried out at a microtubule concentration (2.5 µM) that gave two-fold stimulation of dynein. At increasing concentrations of dynapyrazole-A dynein's microtubule-stimulated ATPase activity saturated at $\sim 0.5 \text{ s}^{-1}$ (*Figure 6E*). Fitting to this non-zero plateau value gave an $IC_{50}$ of dynapyrazole-A of $6.2 \pm 1.6$ µM. In comparison, dose-dependent inhibition of dynein 1's microtubule-stimulated ATPase activity by ciliobrevin D did not reveal a plateau up to the highest concentrations the compound remained soluble in these assay conditions (*Figure 6—figure supplement 2*).

To better understand how dynapyrazole-A inhibits dynein activity, we monitored its effect on ADP-vanadate-dependent photocleavage of dynein. In this assay, an established read-out of nucleotide binding at dynein's AAA1 site (*Lee-Eiford et al., 1986*; *Kon et al., 2004*), ultraviolet irradiation leads to ADP-vanadate-dependent photocleavage of dynein at the AAA1 site, resulting in two protein fragments (*Gibbons and Gibbons, 1987*). Using His-dynein 1, 100 µM ATP, and 100 µM vanadate, we observed $42 \pm 3\%$ photocleavage (n = 3) of His-dynein 1 (*Figure 6F and G*) and fragments consistent with cleavage at AAA1. At a dynapyrazole-A concentration near its $IC_{50}$ for in vitro dynein inhibition (4 µM), ADP-vanadate dependent photocleavage of dynein was reduced ~2.5-fold relative to controls ($17 \pm 2\%$ photocleavage, n = 3). These data are consistent with dynapyrazole-A binding at the AAA1 site.

To further dissect how dynein is inhibited by dynapyrazole, we expressed and purified a construct with a Walker A lysine-to-alanine substitution at AAA3 (K2601A), a mutation that disrupts nucleotide binding at AAA3 (*Kon et al., 2004*; *Silvanovich et al., 2003*). We found that the mutant protein eluted during size exclusion chromatography with a major peak at a similar elution volume to that for the wild-type construct (12.4 mL for the mutant, 12.2 mL for wild-type, *Figure 6H and I*), suggesting that the mutations do not disrupt overall protein folding. We also purified a comparable AAA1 (K1912A) mutant, but this protein had ~80–90% reduced ATPase activity, making analysis of its ATPase activity difficult (*Figure 6—figure supplement 3*). We found that the AAA3 mutant had basal activity that was elevated relative to wild-type ($1.1 \pm 0.2 \text{ s}^{-1}$, n = 5), as has been previously shown for yeast dynein with a similar mutation (*DeWitt et al., 2015*). The ATPase activity of this mutant was stimulated by microtubules to $1.5 \text{ s}^{-1}$ (3 µM microtubules) and $1.7 \text{ s}^{-1}$ (11.5 µM microtubules) (*Figure 6J*). The reduced microtubule stimulation of the ATPase activity is consistent with findings for comparable dynein constructs from other organisms (*DeWitt et al., 2015*; *Kon et al., 2004*).

Across the range of microtubule concentrations tested, dynapyrazole-A (30µM) reduced the AAA3 mutant dynein's ATPase activity by 70–80% (*Figure 6J*). Surprisingly, unlike what we observed for the wild-type protein, strong

**Video 7.** Lysosome dynamics in CAD cell neurites treated with dynapyrazole-A (compound **8**). Time-lapse movies of Lysotracker Red-treated CAD cells in the presence dynapyrazole-A (compound **8**, 5 µM). Time, min:sec.

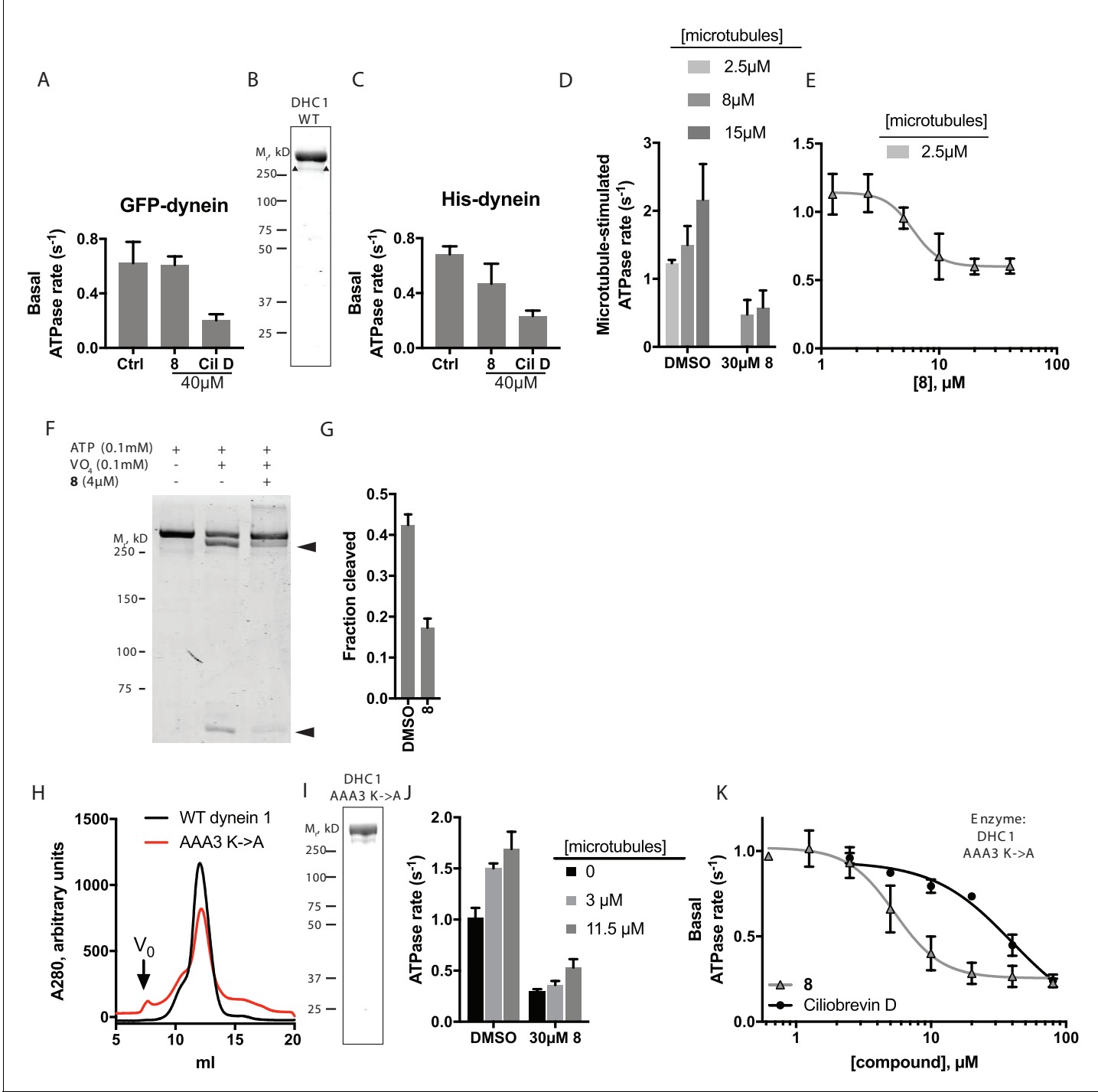

**Figure 6.** Analysis of the mechanism of dynein inhibition by dynapyrazole-A (compound **8**). (**A**) Basal ATPase activity of GFP-dynein in the solvent control (2% DMSO, n = 8) and in the presence of **8** (40µM, n = 4) and ciliobrevin D (40 µM, n = 4). (**B**) SDS-PAGE analysis (Coomassie blue stain) of His-dynein 1, ~0.5 µg protein loaded. Mass spectrometry showed that the impurity (~15%, triangles) is likely to be a fragment of dynein (**Figure 6—figure supplement 1**) (**C**) Basal ATPase activity of His-dynein in the solvent control (2% DMSO, n = 11) and in the presence of **8** (40µM, n = 11) and ciliobrevin D (40 µM, n = 5). (**D**) Microtubule-stimulated ATPase activity of His-dynein 1 across a range of microtubule concentrations in the solvent control (2% DMSO) or in the presence of 30 µM **8** (2.5 µM microtubules, n = 5; 8 µM microtubules, n = 4; 8 µM microtubules, 30 µM **8**, n = 3; 15 µM microtubules, 2% DMSO, n = 4; 15 µM microtubules, 30 µM **8**, n = 3) (**E**) Dose-dependent inhibition of microtubule-stimulated His-dynein 1 ATPase activity by **8** (2.5 µM microtubules, IC$_{50}$: 6.2 ± 1.6 µM, n = 3). (**F**) SDS-PAGE analysis (Coomassie blue stain) of dynein following irradiation with ultraviolet light at 365 nm. The components included in the photocleavage reaction loaded into each lane are indicated above the lane. Arrowheads indicate dynein photocleavage products. (**G**) Analysis of gel band intensity for photocleavage reactions. Values are mean + S.D., n = 3. (**H**) Gel filtration traces

*Figure 6 continued on next page*

*Figure 6 continued*

(Superose 6) for His-dynein 1 wild-type and AAA3 Walker A mutant. Peak elution volumes are 12.2, and 12.4 mL, respectively. $V_o$, void volume. (**I**) SDS-PAGE analysis (Coomassie blue stain) of Walker A mutant His-dynein 1 protein, ~0.5 µg protein loaded. Mass spectrometry data confirming the presence of the K2601A mutation in this construct is shown in *Figure 6—figure supplement 4*. (**J**) Basal and microtubule-stimulated ATPase activity of the AAA3 Walker A-mutant His-dynein 1 in the solvent control (2% DMSO) and in the presence of **8** (30µM). (**K**) Inhibition of the basal ATPase activity of the AAA3 Walker A-mutant His-dynein 1 by **8** (IC$_{50}$: 5.5 ± 1.6 µM, n = 5) and ciliobrevin D (IC$_{50}$: 38.4 ± 6.3 µM, n = 3). IC$_{50}$ values reported reflect the mean (±S.D.) of separate IC$_{50}$ values obtained from independent dose-response analyses. For (**E**) and (**K**), data were fit to a four-parameter sigmoidal dose-response curve in PRISM and fits were constrained such the value at saturating compound >0. All ATPase assays were performed at 1 mM MgATP and 2% DMSO. All data presented are mean ± S.D. of n $\geq$ 3 data points, except in K, where replicate numbers for individual datapoints were as follows. **8**: 80 µM-2, 40 µM-5, 20 µM-5, 10 µM-5, 5 µM-5, 2.5 µM-5, 1.3 µM-5, 0.6 µM-2. Ciliobrevin D: 80 µM-2, 40 µM-3, 20 µM-3, 10 µM-3, 5 µM-3, 2.5 µM-3.

The following figure supplements are available for figure 6:

**Figure supplement 1.** Mass spectrometry-based analysis of wild-type His-dynein 1.

**Figure supplement 2.** Dose-dependent inhibition of microtubule-stimulated His-dynein 1 ATPase activity by ciliobrevin D (2.5 µM microtubules).

**Figure supplement 3.** Purification and testing of His-dynein 1 with Walker A mutation in AAA1.

**Figure supplement 4.** Mass spectrometry-based analysis of His-dynein 1 with Walker A mutation in AAA3.

**Figure supplement 5.** Sequence analysis of human cytoplasmic dynein isoforms 1 and 2.

inhibition of the basal activity of this mutant construct was also observed (~70% inhibition; 30 µM dynapyrazole-A). A dose-dependent analysis showed that dynapyrazole-A inhibits the basal activity of the AAA3 mutant with an IC$_{50}$ of 5.5 ± 1.6 µM, while ciliobrevin D has an IC$_{50}$ of 38 ± 6 µM (*Figure 6K*). Even in the context of the AAA3 mutant enzyme, a residual ATPase activity of ~0.2 s$^{-1}$ was observed at the highest dynapyrazole-A concentrations tested (*Figure 6K*). It is noteworthy that dynapyrazole-A inhibits basal activity of the AAA3 mutant and the microtubule-stimulated ATPase activity of the wild-type enzyme with similar potency, suggesting that a common ATPase site, likely AAA1, might be inhibited in both contexts.

## Discussion

In this study, we analyzed the structure of the ciliobrevins and designed tricyclic derivatives in which the benzoylacrylonitrile of ciliobrevin was replaced with a cyanopyrazole. These compounds inhibited dynein in vitro and in cells more potently than ciliobrevins and have improved chemical properties. We also find that dynapyrazole-A inhibits dynein by a mechanism that is distinct from that of ciliobrevins.

The pyrazoloquinazolinone scaffold of the dynapyrazoles addresses many of the limitations of the ciliobrevins. First, activated acrylonitriles, such as the benzoylacrylonitrile at the core of the ciliobrevin scaffold, have been shown to be susceptible to attack by intracellular nucleophiles, raising the possibility of unwanted reactivity of these dynein inhibitors (*Serafimova et al., 2012*). While ciliobrevins have been observed to be unstable under standard laboratory storage conditions, the pyrazoloquinazolinone of the dynapyrazoles is unlikely to be reactive or unstable but retains many of the electrostatic and steric elements present in ciliobrevin (*Sainath and Gallo, 2015*). Second, the potency of dynapyrazole-A is similar across in vitro and cell-based assays, consistent with dynein being its cellular target. This was not consistently observed for ciliobrevin analogs. Dynapyrazole-A and -B inhibited dynein-dependent microtubule gliding six- to eight-fold more potently than ciliobrevins; however, different mechanisms of inhibition make it difficult to relate this change in potency directly to an increased binding affinity. Third, the ciliobrevin scaffold is present in chemical screening libraries and a close derivative has annotated anti-malarial activity (*Pillai et al., 2010*). In contrast, to our knowledge, the aryl-substituted or cyclopropyl-phenyl-substituted cyano-pyrazoloquinazolinone structure of compounds **4-8** has not previously been reported in a bioactive compound. The hydrophobicity of the dynapyrazoles remains a limitation that needs to be addressed. We note that

the high ClogP value of dynapyrazole-A, which is similar to that of ciliobrevins, may explain why this compound, like ciliobrevin D, has reduced activity in the presence of high protein concentrations (e.g. serum). Likewise, acute reversal of intraflagellar transport inhibition was most effective using cell culture medium with serum. We believe that the dynapyrazole scaffold will be valuable for further optimization of these chemical probes.

In current models, the motion of membrane-bound organelles requires a balance of motors moving toward microtubule plus- and minus-ends (*Barlan et al., 2013*). Studies of different organelles across several cell types have established that depletion or inhibition of dynein reduces bidirectional organelle motion (*Waterman-Storer et al., 1997*; *Gross et al., 2002*; *Martin et al., 1999*). Remarkably, organelle motion following dynein depletion can be restored by add-back of a minus-end directed kinesin, suggesting that micromechanical coupling is involved and an opposing force is needed to activate directional motion of the motor proteins that drive the directional motion of these cellular cargoes (*Barlan et al., 2013*; *Ally et al., 2009*). These observations provide a plausible explanation for the bi-directional inhibition of lysosome motility upon dynapyrazole-A treatment. Although the individual cargos in the cilium are not likely membrane-bound, this intracellular transport is no less complex than organelle movement in the bulk cytoplasm (*Prevo et al., 2017*). We believe that dynapyrazole-A will be a useful tool to dissect how the length and shape of the cilium is linked to directional transport and to tease apart the contributions of active transport and passive diffusion of proteins that participate in Hedgehog signaling.

Dynapyrazole-A blocks both dynein 1 and dynein 2. As dynein 1 plays a critical role in numerous cellular processes, its persistent inhibition is expected to be toxic to cells (*Figure 5—figure supplement 5*). However, acute inhibition may be useful in deciphering dynein function in different contexts. Can dynapyrazoles be chemically modified to develop inhibitors that are selective for dynein 1 or 2? Previously, we found that modest selectivity for inhibition of dynein 2 relative to dynein 1 can be achieved under certain conditions ([ATP] <1 μM) by appending tolyl ethers at the 7-position of the ciliobrevin quinazolinone (*See et al., 2016*). However, inhibition of dynein's ATPase activity by these ciliobrevin analogs is suppressed at higher ATP concentrations (e.g. 100 μM) in vitro. Studies of mammalian dynein 1 indicate that there are two ATP-binding sites with $K_m$ values separated by two orders of magnitude (~2 μM and ~600 μM [*Ross et al., 2006*]). It is possible that the observed selectivity is achieved by targeting the high-affinity ATPase site and therefore, at higher, close to physiologic ATP concentrations, the inhibition by these ciliobrevin analogs is suppressed. In contrast, dynapyrazoles inhibit dynein at high ATP concentrations (1 mM), consistent with a different mode of inhibition. Also, substitution of this compound with a methyl ether at the 6-position abrogated dynein inhibition. These data suggest that the selectivity gains observed from appending tolyl ethers to the ciliobrevin scaffold are unlikely to be applicable to dynapyrazoles and developing a dynein isoform-selective dynapyrazole derivative will require more extensive structure-activity relationship studies.

Of the six AAA sites in dynein, only mutations at AAA1 and AAA3 have been found to block motility (*Kon et al., 2004*). In current models, ATPase activity at AAA1 is linked to rigid-body movements of subdomains during individual steps of the motor protein along microtubule tracks, while nucleotide state at AAA3 modulates the activity at AAA1 (*Carter, 2013*). Three lines of evidence indicate that dynapyrazole-A targets the AAA1 site. First, dynapyrazole-A blocks ADP-vanadate-dependent photocleavage at site 1. Second, dynapyrazole-A inhibits the activity of a dynein 1 construct with a mutation in the AAA3 domain (Walker A lysine). In this construct, the ATPase activity is expected to be mainly due to the AAA1 site. Third, sequence comparisons indicate that the AAA1 sites in dynein 1 and 2 are highly conserved, while the other AAA sites are less conserved. In fact, the residues within 4 Å of the bound nucleotide in AAA1 are identical between dynein 1 and 2 (*Figure 6—figure supplement 5*). This sequence similarity and our observation that dynapyrazole-A inhibits dynein 1 and 2 with comparable potency are consistent with a model in which this compound selectively inhibits hydrolysis of dynein's AAA1 site. However, further biophysical and structural studies will be required to determine whether dynapyrazole-A binds at or near the AAA1 site.

Our finding that dynapyrazole-A does not potently inhibit ATP hydrolysis in the absence of microtubules suggests that, for dynein 1, the AAA1 site contributes to only a small fraction of this enzyme's basal activity. We posit that a separate site, likely AAA3, accounts for the majority of dynein's ATPase activity in the absence of microtubules. This site likely remains active even when ATPase activity at AAA1 is inhibited by dynapyrazoles. Currently, we cannot exclude the possibility

that additional ATPase sites, such as AAA4, may also contribute partially to the basal ATPase activity of dynein. The nucleotide state at AAA3 is known to regulate the ATPase cycle of AAA1 when microtubules are present, with the apo- and ATP-bound states of AAA3 slowing overall hydrolysis and the ADP-bound state of AAA3 leading to rapid hydrolysis at AAA1 in wild-type dynein (*Bhabha et al., 2014*; *DeWitt et al., 2015*). Our finding that the AAA3 mutant has elevated basal activity relative to the wild-type is consistent with the protein being in a state that mimics one in which hydrolysis at AAA1 is activated. Although additional biophysical studies are needed to analyze this further, dynapyrazole-A is likely to be useful as tool for chemical 'separation of function' to distinguish between dynein's microtubule stimulated and basal activities in different contexts.

While restricting conformational flexibility of ligands is a well-established strategy for improving potency, its successful application typically depends on structural data that reveal the compound conformation required to bind its target protein (*Babine and Bender, 1997*). In the case of the ciliobrevins, the geometric constraints we designed were guided by conformational analysis of the compounds alone and afforded ~6–8 fold improvement in potency. We note that replacement of a scaffold prone to isomerization with a fixed tricyclic pyrazoloquinazolinone did not lead directly to improved potency. Rather, incorporation of a cyclopropyl group into the dynapyrazole scaffold was required, likely by better matching the shape of the ciliobrevin pharmacophore. Our unexpected finding that conformational restriction led to changes both in potency and mechanism of inhibition raises the question of whether alternative scaffolds with different conformational constrains could reveal still other modes of dynein inhibition. More broadly, conformational restriction may be an effective strategy for improving the few AAA+ inhibitors described to date. Compounds with distinct mechanisms of action, which may result from these efforts, will be especially valuable for these multi-site enzymes whose activity is often regulated by intricate allosteric communication.

## Materials and methods

### Chemical synthesis

General chemical methods: Solvents and reagents were purchased from VWR or Sigma Aldrich. All reactions involving air- or moisture-sensitive compounds were performed under nitrogen atmosphere using dried glassware. $^{1}$H NMR spectra were recorded at 500MHz on a Bruker Advance III HD 500 MHz NMR spectrometer equipped with a TCI cryogenic probe with enhanced $^{1}$H and $^{13}$C detection. Chemical shifts are reported in parts per million (ppm) and coupling constants (J) are expressed in hertz (Hz). All data were collected at 298K, and internally referenced for $^{1}$H to the residual chloroform signal at 7.26 ppm; to DMSO signal at 2.50 ppm; or to TMS at 0 ppm. Flash chromatography purifications were performed on CombiFlash Rf (Teledyne ISCO) as the stationary phase. The LC-MS was performed on a Waters ACQUITY H-Class UPLC/MS with a PDA eLambda detector and QDa mass spectrometer. Reactions under microwave irradiation were performed on a CEM Explorer 48 System.

### (2*E*)−3-(2,4-Dichlorophenyl)−2-[7-(2-morpholinoethoxy)−4-oxo-1H-quinazolin-2-ylidene]−3-oxo-propanenitrile (1)

**Chemical structure 1.** (2*E*)-3-(2,4-dichlorophenyl)-2-[7-(2-morpholinoethoxy)-4-oxo-1H-quinazolin-2-ylidene]-3-oxo-propanenitrile (1).

To a mixture of (2E)−3-(2,4-dichlorophenyl)−2-(7-hydroxy-4-oxo-1H-quinazolin-2-ylidene)−3-oxo-propanenitrile (82 mg, 219 µmol, synthesized as described previously) (*See et al., 2016*) and

potassium carbonate (48.7 mg, 351 µmol) in dimethylformamide (10 mL) was added 4-(2-chloroethyl) morpholine hydrochloride (37.13 mg, 200 µmol), and the resulting mixture was stirred at 80°C for 18 hr. The mixture was allowed to cool, and the filtrate was concentrated in vacuo. The residue was purified by column chromatography (ethyl acetate/methanol) to yield (2E)−3-(2,4-dichlorophenyl)−2-[7-(2-morpholinoethoxy)−4-oxo-1H-quinazolin-2-ylidene]−3-oxo-propanenitrile (50.20 mg, 103.01 µmol, 52% yield). [1]H NMR (500 MHz, DMSO-$d_6$) δ 13.36 (s, 1 hr), 9.88 (s, 1 hr), 7.95 (d, $J$ = 8.8 Hz, 1 hr), 7.66 (d, $J$ = 2.0 Hz, 1 hr), 7.50–7.41 (m, 2 hr), 7.05–6.95 (m, 1 hr), 6.94–6.86 (m, 1 hr), 4.50 (s, 2 hr), 4.09–3.93 (m, 2 hr), 3.68 (d, $J$ = 33.0 Hz, 2 hr), 3.57 (s, 2 hr), 3.24 (s, 2 hr). LCMS m/z: 487.1 [M+H]$^+$.

**Scheme 1.** Synthesis of **2** based on literature precedent (*Süsse and Johne, 1981*).

## 2-[1-(2,4-Dichlorophenyl)ethylidene]propanedinitrile

**Chemical structure 2.** 2-[1-(2,4-dichlorophenyl)ethylidene]propanedinitrile.

To a solution of 1-(2,4-dichlorophenyl)ethanone (1 g, 5.29 mmol) in toluene (20 mL) - acetic acid (2 mL) were added malononitrile (349.45 mg, 5.29 mmol) and ammonium acetate (407 mg, 5.29 mmol) at room temperature. The mixture was stirred at 100°C for 13 hr. The mixture was neutralized with sat. NaHCO$_3$ (aq.) and extracted with ethyl acetate. The combined organic layer was washed with water and brine, dried over MgSO$_4$, filtered and concentrated in vacuo. The residue was purified by column chromatography (hexane/ethyl acetate) to yield 2-[1-(2,4-dichlorophenyl)ethylidene]propane-dinitrile (907 mg, 3.83 mmol, 72% yield) as a colorless oil. [1]H NMR (500 MHz, Chloroform-d) δ 7.57 (s, 1 hr), 7.43 (d, J = 8.3 Hz, 1 hr), 7.19 (d, J = 8.3 Hz, 1 hr), 2.62 (s, 3 hr). LCMS m/z: 235.1 [M-H]$^-$.

## 2-[2-Bromo-1-(2,4-dichlorophenyl)ethylidene]propanedinitrile

**Chemical structure 3.** 2-[2-bromo-1-(2,4-dichlorophenyl)ethylidene]propanedinitrile.

To a solution of 2-[1-(2,4-dichlorophenyl)ethylidene]propanedinitrile (310.50 mg, 1.31 mmol) in carbon tetrachloride (4 mL) was added bromine (230 mg, 1.44 mmol) at room temperature. The mixture was stirred at room temperature for 15 min and at 70°C for 8.5 hr. The starting material remained,

so bromine (620 mg, 200 µL, 3.88 mmol) was added. The mixture was stirred at 70°C for 21 hr. The mixture was diluted with sat. NaS$_2$O$_3$ (aq.) and extracted with dichloromethane. The combined organic layer was washed with sat. NaS$_2$O$_3$ (aq.) and brine, dried over MgSO$_4$, filtered and concentrated in vacuo. The residue was purified by column chromatography (hexane/ethyl acetate) to yield 2-[2-bromo-1-(2,4-dichlorophenyl)ethylidene]propanedinitrile (157 mg, 497 µmol, 38% yield) as a colorless oil. [1]H NMR (500 MHz, Chloroform-*d*) δ7.59 (s, 1 hr), 7.47 (d, *J* = 8.3 Hz, 1 hr), 7.34 (d, *J* = 8.3 Hz, 1 hr), 4.58 (s, 2 hr). LCMS m/z: 314.856 [M-H]⁻.

## 2-(2,4-Dichlorophenyl)—5-oxo-4H-pyrrolo[1,2-a]quinazoline-3-carbonitrile (2)

**Chemical structure 4.** 2-(2,4-dichlorophenyl)-5-oxo-4H-pyrrolo[1,2-a]quinazoline-3-carbonitrile (2).

To a solution of 2-[2-bromo-1-(2,4-dichlorophenyl)ethylidene]propanedinitrile (73.70 mg, 233.24 µmol) in isopropyl alcohol (30 mL) was added methyl 2-aminobenzoate (70.51 mg, 466.48 µmol) at room temperature. The mixture was stirred at 100°C for 20 hr. The mixture was concentrated. The residue was purified by column chromatography (hexane/ethyl acetate) to give 2-(2,4-dichlorophenyl)—5-oxo-4H-pyrrolo[1,2-a]quinazoline-3-carbonitrile (12.4 mg, 35 µmol, 15% yield) as a colorless solid. [1]H NMR (500 MHz, Chloroform-*d*) δ 8.81 (s, 1 hr), 8.40 (d, *J* = 7.8 Hz, 1 hr), 7.84 (t, *J* = 7.8 Hz, 1 hr), 7.66 (d, *J* = 8.3 Hz, 1 hr), 7.57–7.46 (m, 3 hr), 7.40 (s, 1 hr), 7.36 (d, *J* = 8.2 Hz, 1 hr). LCMS m/z: 354.1 [M+H]⁺.

## 2-(2,4-Dichlorophenyl)—4H-pyrazolo[1,5-a]quinazolin-5-one (3)

**Chemical structure 5.** 2-(2,4-dichlorophenyl)-4H-pyrazolo[1,5-a]quinazolin-5-one (3).

A mixture of 2-hydrazinobenzoic acid (43.32 mg, 284.7 µmol), 3-(2,4-dichlorophenyl)—3-oxo-propanenitrile (41.20 mg, 192.48 µmol) and acetic acid (2 mL) was stirred at 150°C under microwave irradiation for 30 min. The mixture was diluted with water and ethyl acetate, the insoluble material was collected by filtration to give the desired compound as a colorless solid (7.5 mg). The filtrate was extracted with ethyl acetate. The combined organic layer was washed with water and brine, dried over MgSO4, filtered and concentrated in vacuo. The solid was washed with ethyl acetate to give the desired compound (9.4 mg). The combined solid was washed with hexane to yield 2-(2,4-dichlorophenyl)—4H-pyrazolo[1,5-a]quinazolin-5-one (16.4 mg, 49.7 µmol, 26% yield) as a colorless solid.[1]H NMR (500 MHz, DMSO-d6) δ 12.34 (s, 1 hr), 8.19 (dd, J = 11.4, 8.2 Hz, 2 hr), 7.99 (d, J = 8.4 Hz, 1 hr), 7.94 (t, J = 7.7 Hz, 1 hr), 7.80 (s, 1 hr), 7.61–7.53 (m, 2 hr), 6.42 (s, 1 hr). LCMS m/z: 329.9 [M+H]⁺.

**Scheme 2.** Synthesis of aminopyrazole precursor for **4** based on literature precedent for related compounds (**Bussenius et al., 2012**).

## 2-[(2,4-Dichlorophenyl)-hydroxy-methylene]propanedinitrile

**Chemical structure 6.** 2-[(2,4-dichlorophenyl)-hydroxy-methylene]propanedinitrile.

To a stirred solution of propanedinitrile (4.71 g, 71.33 mmol) in THF (150 mL), sodium hydride (5.71 g, 142.66 mmol, 60% purity) was added at 0°C. The mixture was stirred at room temperature for 1 hr. The solution of 2,4-dichlorobenzoyl chloride (14.62 g, 69.8 mmol, 10 mL) in THF (50 mL) was added to the mixture at 0°C. The mixture was stirred at room temperature for 4 hr. The mixture was diluted with 1N HCl (aq.) and extracted with ethyl acetate. The combined organic layer was washed with water and brine, dried over MgSO$_4$, filtered and concentrated in vacuo. The residue was purified by column chromatography (ethyl acetate/hexanes) to yield 2-[(2,4-dichlorophenyl)-hydroxy-methylene]propanedinitrile (17.38 g, 72.70 mmol, 104% yield) as light yellow amorphous powder. This product was subjected to the next reaction. [1]H NMR (500 MHz, Chloroform-*d*) δ7.55 (s, 1 hr), 7.46–7.38 (m, 2 hr). LCMS m/z: 236.992 [M-H]⁻.

## 2-[(2,4-Dichlorophenyl)-methoxy-methylene]propanedinitrile

**Chemical structure 7.** 2-[(2,4-dichlorophenyl)-methoxy-methylene]propanedinitrile.

A mixture of 2-[(2,4-dichlorophenyl)-hydroxy-methylene]propanedinitrile (17.38 g, 72.70 mmol), dimethyl sulfate (18.34 g, 145.40 mmol, 13.79 mL) and N-ethyl-N-isopropyl-propan-2-amine (28.19 g, 218.10 mmol, 38.09 mL) in dioxane (200 mL) was stirred at 60°C for 23 hr. The reaction was cooled to room temperature and concentrated in vacuo. The residue was dissolved with ethyl acetate and quenched with water. The organic layer was separated, washed with brine, dried over MgSO$_4$, filtered and concentrated in vacuo. The residue was purified by column chromatography (hexane/ethyl acetate) to yield 2-[(2,4-dichlorophenyl)-methoxy-methylene]propanedinitrile (3.19 g, 12.60 mmol, 17% yield) as a brown solid. (Known compound, cas: 1188083-55-7). [1]H NMR (500 MHz, Chloroform-*d*) δ 7.62 (d, *J* = 1.9 Hz, 1 hr), 7.49 (dd, *J* = 8.3, 1.9 Hz, 1 hr), 7.34 (d, *J* = 8.3 Hz, 1 hr), 3.85 (s, 3 hr).

## 5-Amino-3-(2,4-dichlorophenyl)−1H-pyrazole-4-carbonitrile

**Chemical structure 8.** 5-amino-3-(2,4-dichlorophenyl)-1H-pyrazole-4-carbonitrile.

The mixture of 2-[(2,4-dichlorophenyl)-methoxy-methylene]propanedinitrile (3.19 g, 12.60 mmol) and hydrazine hydrate (694 mg, 13.86 mmol, 672 µL) in ethanol (50mL) was stirred at 80°C for 4.5 hr. Additional hydrazine hydrate (252.30 mg, 5.04 mmol, 244.48 uL) was added to the mixture and it was stirred at 80°C for 1.5 hr. The reaction was concentrated in vacuo. The residue was washed with ethanol to yield 5-amino-3-(2,4-dichlorophenyl)−1H-pyrazole-4-carbonitrile (1.73 g, 6.84 mmol, 54% yield) as an off-white solid. [1]H NMR: (500MHz, DMSO-$d_6$) δ 12.31 (s, 1 hr), 7.76 (s, 1 hr), 7.58–7.41 (m, 2 hr), 6.49 (s, 2 hr). LCMS m/z: 253.184 [M+H]$^+$.

## 2-(2,4-Dichlorophenyl)−5-oxo-4H-pyrazolo[1,5-a]quinazoline-3-carbonitrile, 4

**Chemical structure 9.** 2-(2,4-dichlorophenyl)-5-oxo-4H-pyrazolo[1,5-a]quinazoline-3-carbonitrile, 4.

A mixture of 5-amino-3-(2,4-dichlorophenyl)−1H-pyrazole-4-carbonitrile (100 mg, 395.12 µmol), dipotassium carbonate (81.91 mg, 592.68 µmol) and methyl 2-fluorobenzoate (73.08 mg, 474.14 µmol, 60.40 µL) in dimethylformamide (1 mL) was stirred at 140°C for 30 min. The mixture was poured into water, and extracted with ethyl acetate. The organic layer was washed with water and brine respectively, dried over MgSO$_4$ and concentrated in vacuo. The residue was purified by silica-gel column chromatography (hexane/ethyl acetate) to yield 2-(2,4-dichlorophenyl)−5-oxo-4H-pyra-zolo[1,5-a]quinazoline-3-carbonitrile (14.2 mg, 39.98 µmol, 10% yield) as a white solid. [1]H NMR: (500MHz, DMSO-$d_6$) δ 8.09 (d, J = 7.8 Hz, 1 hr), 7.94 (d, J = 8.2 Hz 1 hr), 7.83 (d, J = 2.0 Hz 1 hr), 7.73–7.56 (m, 3 hr), 7.40 (t, J = 7.5 Hz, 1 hr). LCMS m/z: 353.136 [M-H]$^-$.

**Scheme 3.** Synthesis of cyclopropyl-substituted aminopyrazole precursor for 5, 6, 7, and 8.

## 2-((1-(4-Chlorophenyl)cyclopropyl) (hydroxy)methylene)malononitrile

**Chemical structure 10.** 2-((1-(4-chlorophenyl)cyclopropyl)(hydroxy)methylene)malononitrile.

To a solution of 1-(4-chlorophenyl)cyclopropanecarboxylic acid (**1**, 10 g, 50.86 mmol) in THF (100 mL) were added oxalyl chloride (7.75 g, 61 mmol, 5.33 mL) and dimethylformamide (37.17 mg, 508.60 µmol, 39.54 uL). The mixture was stirred at room temperature for 0.5 hr. The mixture was concentrated in vacuo. The mixture was added to a solution of propanedinitrile (3.36 g, 50.86 mmol) and sodium hydride (4.07 g, 101.72 mmol, 60% purity) in THF (100 mL) at 0°C. The mixture was stirred at room temperature for 1 hr. The mixture was diluted with 1N HCl and extracted with ethyl acetate. The combined organic layer was washed with water and brine, dried over $Na_2SO_4$, filtered and concentrated under reduced pressure. The residue was purified by column chromatography (hexane/ethyl acetate) to yield 2-((1-(4-chlorophenyl)cyclopropyl) (hydroxy)methylene)malononitrile (12 g, 49 mmol, 96% yield) as a light yellow oil. [1]H NMR (500 MHz, DMSO-$d_6$) δ 7.36–7.30 (m, 2 hr), 7.22 (dd, $J$ = 7.8, 2.0 Hz, 2 hr), 1.22–1.16 (m, 2 hr), 0.97 (d, $J$ = 3.6 Hz, 2 hr). LCMS m/z: not detected.

## 2-((1-(4-Chlorophenyl)cyclopropyl) (methoxy)methylene)malononitrile

**Chemical structure 11.** 2-((1-(4-chlorophenyl)cyclopropyl)(methoxy)methylene)malononitrile.

To a solution of 2-[1-(4-chlorophenyl)cyclopropanecarbonyl]propanedinitrile (**2**, 12 g, 49 mmol) in dioxane (200 mL) and $H_2O$ (20 mL) were added dimethyl sulfate (18.56 g, 147.1 mmol, 13.9 mL) and NaHCO$_3$ (20.60 g, 245.20 mmol). The mixture was stirred at 100°C for 5 hr. The mixture was poured into water and extracted with ethyl acetate. The organic layer was washed with brine, dried over $Na_2SO_4$, and concentrated in vacuo. The residue was purified by column chromatography (hexane/ethyl acetate) to yield 2-((1-(4-chlorophenyl)cyclopropyl)(methoxy)methylene)malononitrile (3.38 g, 13.1 mmol, 27% yield) as a yellow oil. [1]H NMR (500 MHz, DMSO-$d_6$) δ 7.55–7.45 (m, 2 hr), 7.33–7.22 (m, 2 hr), 4.04 (s, 3 hr), 1.80–1.71 (m, 2 hr), 1.68–1.61 (m, 2 hr). LCMS m/z: not detected.

## 5-Amino-3-[1-(4-chlorophenyl)cyclopropyl]−1H-pyrazole-4-carbonitrile

**Chemical structure 12.** 5-Amino-3-[1-(4-chlorophenyl)cyclopropyl]-1H-pyrazole-4-carbonitrile.

To a solution of 2-[[1-(4-chlorophenyl)cyclopropyl]-methoxy-methylene]propanedinitrile (**3**, 3.38 g, 13.07 mmol) in ethanol (100 mL) was added hydrazine hydrate (981.43 mg, 19.61 mmol). The mixture was stirred at 80°C for 2 hr. The mixture was concentrated in vacuo. The residue was purified by column chromatography (hexane/ethyl acetate) to yield 5-amino-3-[1-(4-chlorophenyl)cyclopropyl]−1H-

pyrazole-4-carbonitrile (2.77 g, 10.71 mmol, 82% yield) as a white powder. [1]H NMR (500 MHz, DMSO-$d_6$) δ 11.73 (s, 1 hr), 7.35 (d, $J$ = 8.0 Hz, 2 hr), 7.27–7.16 (m, 2 hr), 6.28 (s, 2 hr), 1.43–1.13 (m, 4 hr). LCMS m/z: 259 [M+H]$^+$.

## 2-[1-(4-Chlorophenyl)cyclopropyl]—5-oxo-4H-pyrazolo[1,5-a]quinazoline-3-carbonitrile (5)

**Chemical structure 13.** 2-[1-(4-chlorophenyl)cyclopropyl]-5-oxo-4H-pyrazolo[1,5-a]quinazoline-3-carbonitrile (5).

A mixture of 5-amino-3-[1-(4-chlorophenyl)cyclopropyl]—1H-pyrazole-4-carbonitrile (100 mg, 386.53 µmol), dipotassium carbonate (80.13 mg, 579.80 µmol) and methyl 2-fluorobenzoate (71.50 mg, 463.84 µmol, 59.09 uL) in dimethylformamide (1 mL) was stirred at 140°C for 30 min. The mixture was poured into water, and extracted with ethyl acetate. The organic layer was washed with water and brine respectively, dried over MgSO$_4$ and concentrated in vacuo. The residue was purified by flash chromatography (ethyl acetate/hexane) to yield 2-[1-(4-chlorophenyl)cyclopropyl]—5-oxo-4H-pyrazolo[1,5-a]quinazoline-3-carbonitrile (16 mg, 44 µmol, 11% yield) as a white solid. [1]H NMR (500 MHz, DMSO-$d_6$) δ 13.23 (s, 1 hr), 8.17 (d, $J$ = 7.9 Hz, 1 hr), 8.10 (d, $J$ = 8.2 Hz, 1 hr), 7.95 (t, $J$ = 7.8 Hz, 1 hr), 7.59 (t, $J$ = 7.6 Hz, 1 hr), 7.38 (d, $J$ = 8.2 Hz, 2 hr), 7.32 (d, $J$ = 8.3 Hz, 2 hr), 1.55 (q, $J$ = 4.6 Hz, 2 hr), 1.42 (q, $J$ = 4.6 Hz, 2 hr). LCMS m/z: 361.233 [M+H]$^+$.

## 2-(1-(4-Chlorophenyl)cyclopropyl)—5-oxo-7-(trifluoromethyl)—4,5-dihydropyrazolo[1,5-a]quinazoline-3-carbonitrile (6)

**Chemical structure 14.** 2-(1-(4-chlorophenyl)cyclopropyl)-5-oxo-7-(trifluoromethyl)-4,5-dihydropyrazolo[1,5-a]quinazoline-3-carbonitrile (6).

A mixture of 5-amino-3-[1-(4-chlorophenyl)cyclopropyl]—1H-pyrazole-4-carbonitrile (80 mg, 309.23 µmol) , methyl 2-fluoro-5-(trifluoromethyl)benzoate (75.56 mg, 340.15 µmol) and dipotassium;carbonate (64.11 mg, 463.85 µmol) in dimethylformamide (1 mL) was stirred at 140°C for 30 min under microwave irradiation (*Deshaies, 2014*). The mixture was poured into water, and extracted with ethyl acetate. The organic layer was washed successively with water and brine, dried over MgSO$_4$ and concentrated in vacuo. The residue was purified by column chromatography (ethyl acetate/hexane) to give a mixture. The amorphous material was triturated with acetonitrile and the white precipitate was collected to yield 2-[1-(4-chlorophenyl)cyclopropyl]—5-oxo-7-(trifluoromethyl)—4H-pyrazolo[1,5-a]quinazoline-3-carbonitrile (26.20 mg, 61.10 µmol, 19.76% yield) as a white solid. [1]H NMR (500 MHz, DMSO-$d_6$) δ 13.49 (s, 1 hr), 8.36 (s, 1 hr), 8.25 (d, $J$ = 1.3 Hz, 2 hr), 7.42–7.35 (m, 2 hr), 7.35–7.27 (m, 2 hr), 1.63–1.52 (m, 2 hr), 1.48–1.38 (m, 2 hr). LCMS m/z: 429.2 [M+H]$^+$.

## 2-[1-(4-Chlorophenyl)cyclopropyl]—7-methoxy-5-oxo-4H-pyrazolo[1,5-a]quinazoline-3-carbonitrile (7)

**Chemical structure 15.** 2-[1-(4-chlorophenyl)cyclopropyl]-7-methoxy-5-oxo-4H-pyrazolo[1,5-a]quinazoline-3-carbonitrile (7).

A mixture of 5-amino-3-[1-(4-chlorophenyl)cyclopropyl]—1H-pyrazole-4-carbonitrile (127.71 mg, 493.65 µmol), dipotassium;carbonate (102.34 mg, 740.47 µmol) and methyl 2-fluoro-5-methoxy-benzoate (100 mg, 543.01 µmol) in dimethylformamide (3 mL) was stirred at 140°C for 40 hr. The mixture was poured into water, and extracted with ethyl acetate. The organic layer was washed with water and brine respectively, dried over MgSO$_4$ and concentrated in vacuo. The residue was purified column chromatography (ethyl acetate/hexane) to yield 2-[1-(4-chlorophenyl)cyclopropyl]—7-methoxy-5-oxo-4H-pyrazolo[1,5-a]quinazoline-3-carbonitrile (9 mg, 24 µmol, 5% yield) as a white solid. [1]H NMR (500 MHz, DMSO-$d_6$) δ 13.24 (s, 1 hr), 8.03 (d, J = 9.0 Hz, 1 hr), 7.61–7.48 (m, 2 hr), 7.39–7.23 (m, 4 hr), 3.89 (s, 3 hr), 1.52 (q, J = 4.5 Hz, 2 hr), 1.39 (q, J = 4.5 Hz, 2 hr). LCMS m/z: 391.30 [M+H]$^+$.

## 2-[1-(4-Chlorophenyl)cyclopropyl]—7-iodo-5-oxo-4H-pyrazolo[1,5-a]quinazoline-3-carbonitrile (8)

**Chemical structure 16.** 2-[1-(4-chlorophenyl)cyclopropyl]-7-iodo-5-oxo-4H-pyrazolo[1,5-a]quinazoline-3-carbonitrile (8).

A mixture of 5-amino-3-[1-(4-chlorophenyl)cyclopropyl]—1H-pyrazole-4-carbonitrile (498.89 mg, 1.93 mmol), methyl 2-fluoro-5-iodo-benzoate (600 mg, 2.14 mmol) and dipotassium carbonate (444.20 mg, 3.21 mmol) in dimethylformamide (10 mL) was stirred at 150°C for 1 hr under microwave irradiation. The reaction was cooled to room temperature and poured into water. The white precipitate was collected and washed with ethyl acetate to yield 2-[1-(4-chlorophenyl)cyclopropyl]—7-iodo-5-oxo-4H-pyrazolo[1,5-a]quinazoline-3-carbonitrile (400 mg, 822 µmol, 38% yield) as a white solid. [1]H NMR (500 MHz, Chloroform-d) δ 9.78 (s, 1 hr), 8.65 (d, J = 2.0 Hz, 1 hr), 8.15 (dd, J = 8.7, 2.0 Hz, 1 hr), 7.91 (d, J = 8.6 Hz, 1 hr), 7.43–7.32 (m, 4 hr), 1.64 (d, J = 2.4 Hz, 2 hr, overlaps with a peak for residual water), 1.41 (q, J = 4.6 Hz, 2 hr). LCMS m/z: 487.0 [M+H]$^+$.

**Scheme 4.** Synthesis of **9**.

## 2-(Chloromethyl)−6-(trifluoromethyl)−2,3-dihydroquinazolin-4(1*H*)-one

**Chemical structure 17.** 2-(chloromethyl)-6-(trifluoromethyl)-2,3-dihydroquinazolin-4(1*H*)-one.

To a solution of sodium methoxide (0.5 M, 2.44 mL) was added 2-chloroacetonitrile (423 mg, 5.6 mmol). The mixture was stirred at 25°C for 30 min. A solution of 2-amino-5-(trifluoromethyl)benzoic acid (1 g, 4.9 mmol) in MeOH (7 mL) was added to the mixture. The mixture was stirred at room temperature for 12 hr. The resulting solid was collected by filtration to yield 2-(chloromethyl)−6-(trifluoromethyl)−1H-quinazolin-4-one (422 mg, 1.61 mmol, 33% yield) as a white solid. [1]H NMR (500 MHz, DMSO-d6) δ 12.98 (s, 1 hr), 8.38 (s, 1 hr), 8.17 (d, J = 8.4 Hz, 1 hr), 7.90 (dd, J = 8.6, 2.4 Hz, 1 hr), 4.61 (d, J = 2.6 Hz, 2 hr). LCMS m/z: 263.03 [M+H]$^{+}$.

## 2-(4-Oxo-6-(trifluoromethyl)−1,4-dihydroquinazolin-2-yl)acetonitrile

**Chemical structure 18.** 2-(4-oxo-6-(trifluoromethyl)-1,4-dihydroquinazolin-2-yl)acetonitrile.

To a solution of 2-(chloromethyl)−6-(trifluoromethyl)−1H-quinazolin-4-one (410 mg, 1.56 mmol) in DMSO (5 mL) was added sodium cyanide (153 mg, 3.12 mmol). The mixture was stirred at 25°C for 3 hr. The mixture was poured into aqueous ammonium chloride, and extracted with ethyl acetate. The organic layer was washed with brine, dried over Na$_2$SO$_4$ and concentrated in vacuo. The residue was purified by column chromatography (ethyl acetate/hexane) to yield 2-[4-oxo-6-(trifluoromethyl)−1H-quinazolin-2-yl]acetonitrile (145 mg, 573 μmol, 37% yield) as a pale yellow solid. [1]H NMR (500 MHz, DMSO-d6) δ 12.83 (s, 1 hr), 8.37 (s, 1 hr), 8.16 (d, J = 8.6 Hz, 1 hr), 7.93–7.86 (m, 1 hr), 4.26 (d, J = 2.6 Hz, 2 hr). LCMS m/z: 254.13 [M+H]$^{+}$.

(*E*)−3-(2,4-dichlorophenyl)−3-oxo-2-(4-oxo-6-(trifluoromethyl)−3,4-dihydroquinazolin-2(1*H*)-ylidene)propanenitrile (9)

**Chemical structure 19.** (*E*)-3-(2,4-dichlorophenyl)-3-oxo-2-(4-oxo-6-(trifluoromethyl)-3,4-dihydroquinazolin-2(1*H*)-ylidene)propanenitrile (9).

To a mixture of 2-[4-oxo-6-(trifluoromethyl)−3H-quinazolin-2-yl]acetonitrile (145 mg, 573 µmol)in dioxane (4 mL) were added 2,4-dichlorobenzoyl chloride (144 mg, 687 µmol) and triethylamine (64 mg, 630 µmol) at 100°C. The mixture was stirred for 15 hr at 100°C. The mixture was poured into water, and extracted with EtOAc. The organic layer was washed with brine, dried over $Na_2SO_4$ and concentrated in vacuo. The resulting solid was collected by filtration using ethyl acetate to yield (2E)−3-(2,4-dichlorophenyl)−3-oxo-2-[4-oxo-6-(trifluoromethyl)−1H-quinazolin-2-ylidene]propanenitrile (159 mg, 373 µmol, 65% yield) as a pale yellow solid. [1]H NMR (500 MHz, DMSO-d6) δ 13.66 (s, 1 hr), 8.23 (s, 1 hr), 7.91 (d, J = 8.5 Hz, 1 hr), 7.67 (q, J = 1.9 Hz, 1 hr), 7.56 (d, J = 8.7 Hz, 1 hr), 7.47 (tdd, J = 8.8, 5.8, 1.7 Hz, 2 hr). LCMS m/z: 428.08 [M+H]$^+$.

## X-ray crystallography of compounds 1 and 5

For compound **1**, measurements were made on a Rigaku R-AXIS RAPID-191R diffractometer using graphite monochromated Cu-K irradiation. The structure was solved by direct methods with SIR2008 (*Burla et al., 2007*) and was refined using full-matrix least-squares on $F^2$ with SHELXL-2013 (*Sheldrick, 2008*). All non-H atoms were refined with anisotropic displacement parameters. The solvent area was disordered, no satisfactory model could be refined. This disordered density was taken into account with the SQUEEZE procedure, as implemented in PLATON (*Spek, 2009*). For compound **5**, measurements were made on a Rigaku XtaLAB P200 diffractometer using graphite monochromated Cu-Kα radiation. The structure was solved by direct methods with SIR2008 (*Burla et al., 2007*) and was refined using full-matrix least-squares on $F^2$ with SHELXL-2014/7 (*Sheldrick, 2008*). All non-H atoms were refined with anisotropic displacement parameters. Supplementary crystallographic data for compounds **1** and **5** have been deposited in the Cambridge Crystallographic Data Centre (CCDC) with accession numbers CCDC 1510769 (1) and CCDC 1510770 (5), respectively. These data can be obtained free of charge from the CCDC via the internet: http://www.ccdc.cam.ac.uk/Community/Requestastructure/Pages/DataRequest.aspx?.

**Chemical structure 20.** ORTEP drawing of 1, thermal ellipsoids are drawn at 20% probability. Color code: black: carbon; blue: nitrogen; yellow: chlorine; red: oxygen.

*Crystal data for* **1**: $C_{23}H_{20}Cl_2N_4O_4$, MW = 487.34; crystal size, 0.21 × 0.10×0.07 mm; colorless, block; monoclinic, space group $P2_1/c$, a = 14.2322 (6) Å, b = 6.7917 (3) Å, c = 25.2715 (18) Å, α = γ = 90°, β = 90.494 (6)°, V = 2442.7 (2) Å$^3$, Z = 4, Dx = 1.325 g/cm$^3$, T = 100 K, μ = 2.698 mm$^{-1}$, λ = 1.54187 Å, $R_1$ = 0.098, $wR_2$ = 0.201.

**Chemical structure 21.** ORTEP of **5**, thermal ellipsoids are drawn at 30% probability. Color code: black: carbon; blue: nitrogen; yellow: chlorine; red: oxygen.

*Crystal data for* **5**: $C_{20}H_{13}ClN_4O$, *MW* = 360.80; crystal size, 0.23 × 0.05×0.03 mm; colourless, needle; monoclinic, space group P2$_1$/c, *a* = 10.32080 (19) Å, *b* = 4.91042 (7) Å, *c* = 34.5332 (5) Å, $\beta$ = 90.6463 (16)°, $\alpha$ = $\gamma$ = 90°, *V* = 1750.01 (5) Å$^3$, *Z* = 4, *Dx* = 1.369 g/cm$^3$, *T* = 100 K, $\mu$ = 2.067 mm$^{-1}$, $\lambda$ = 1.54187 Å, $R_1$ = 0.037, $wR_2$ = 0.096.

## Protein expression and purification

### GFP-human cytoplasmic dynein 2

(AA 1091–4307, uniprot: Q8NCM8), which bears an N-terminal protein A tag, followed by tobacco etch virus (TEV) protease-cleavable linker, followed by a GFP tag, was expressed using the baculovirus/insect cell expression system and purified using a procedure similar to that described previously (*Schmidt et al., 2015*). Briefly, an Sf9 insect cell pellet was resuspended in lysis buffer (30 mM HEPES pH with KOH to 7.4, 50 mM KOAc, 2 mM MgOAc$_2$, 0.2 mM EGTA, 10% [v/v] glycerol, 300 mM KCl, 0.2 mM ATP, 1 mM DTT, 2 mM PMSF, and protease inhibitor cocktail [cOmplete, Roche]) and lysed using a dounce homonizer. The lysate was clarified by centrifugation at 1.2x10$^5$ *g* and then incubated with IgG-sepharose beads beads for 90 min at 4°. The beads were washed with 12 bed volumes of lysis buffer and 12 bed volumes of buffer B (50 mM Tris HCl pH 8.0, 150 mM KOAc, 2 mM MgOAc$_2$, 1 mM EGTA, 10% v/v glycerol, 1 mM DTT, and 0.2 mM ATP). TEV protease was added to the bead slurry and incubated at 4° for 8 hr with rotation. Cleaved soluble protein was collected, concentrated (Ultra-4, Amicon), and loaded onto a gel filtration chromatography column (Superose 6, GE Healthcare). The protein was eluted in buffer C (20 mM Tris HCl pH 8.0, 100 mM KOAc, 2 mM MgOAc$_2$, 1 mM EGTA, 10% v/v glycerol, 1 mM DTT, 50 µM ATP), peak fractions were collected, concentrated, aliquoted, and then frozen in liquid nitrogen.

### GFP-human cytoplasmic dynein 1

(AA 1320–4646 with additional valine [at position 4647], uniprot Q14204), which, like the GFP-dynein 2 construct, bears N-terminal protein A tag, followed by a TEV-cleavable linker, followed by a GFP tag, was expressed using the baculovirus/insect cell expression system and purified using a similar procedure to that described for GFP-dynein 2 with the following modifications:

- The lysis buffer was: 30 mM HEPES (pH 7.5 with KOH), 200 mM NaCl, 1 mM DTT, 1 mM PMSF, protease inhibitor cocktail (Halt$^{TM}$, Thermo).
- Lysis was accomplished by addition of 0.2% triton X-100 followed by incubation on ice for 15 min.
- The following buffer was used in place of buffers B and C: 50 mM Tris HCl (pH 7.8),150 mM KOAc, 2 mM Mg(OAc)$_2$, 1 mM EGTA, 1 mM EDTA, 10% glycerol, 0.1 mM ATP, 1 mM DTT.
- TEV protease was incubated with the bead slurry for 90 min on wet ice without agitation.
- The protein was not concentrated.

### 6x-His-tagged human cytoplasmic dynein 1

(wild-type, K1912A, K2601A, AA 1320–4646, uniprot: Q14204), proteins which contain N-terminal hexahistixine (6x-His), followed by tobacco etch virus (TEV) protease-cleavable linker were expressed using the baculovirus/insect cell expression system and purified as follows. An Sf9 cell pellet was resuspended in a buffer containing, 30 mM HEPES pH 7.6, 200 mM NaCl, 10 mM imidazole, 1 mM

TCEP, 2 mM PMSF, and protease inhibitor cocktails (HALT, Thermo-Fisher and cOmplete, Roche) and lysed by the addition of Triton X-100 to a final concentration of 0.2%. The lysate was clarified by centrifugation at $1.2 \times 10^5$ $g$ and then incubated with Ni-NTA beads for 2 hr at 4°C. The beads were washed with 75 bed volumes of lysis buffer and bound proteins were eluted with buffer containing 30 mM HEPES pH 7.5, 100 mM NaCl, 500 mM imidazole, and 1 mM TCEP. Eluate fractions containing protein were diluted into buffer B (50 mM Tris HCl pH7.8, 150 mM KOAc, 2 mM Mg(OAc)$_2$,1 mM EGTA, 1 mM EDTA, 0.1 mM ATP, 1 mM DTT) and loaded onto a Mono Q anion exchange column (GE Healthcare). The column was eluted using a salt gradient from 150 to 750 mM KOAc over 20 column volumes. Peak fractions were concentrated using a centrifugal concentrator (Amicon) and subjected to size exclusion chromatography on a Superose 6 column using buffer B. For K1912A and K2601A proteins, the ion exchange chromatography step was omitted and NiNTA-eluate fractions were pooled and subjected directly to size exclusion chromatography. Gel filtration revealed a monodisperse peak at an elution volume of 12.2 mL (wild-type), 12.3 mL (K1912A), and 12.4 mL (K2601A), consistent with the expected molecular weight. Peak fractions were pooled, supplemented with glycerol to a final concentration of 20%, and snap frozen. The protein yield was ~1–2 mg of dynein 1 wild-type or mutants per 1L of Sf9 culture. Cleavage of the TEV-cleavable linker following binding to Ni-NTA beads led to protein precipitation.

## Microtubule motility assays

*Reaction setup:* A coverslip-based motility assay was adapted from similar assays reported previously (*Firestone et al., 2012*; *Schmidt et al., 2015*). A flow chamber assembled on a pre-cleaned microscope coverslide (3–5 µL chamber volume) was hydrated with buffer A (25 mM PIPES, 30 mM KCl, 5 mM MgCl$_2$, 1 mM EGTA, 0.01% Triton X-100, 1 mM DTT, 20 µM taxol, pH 7.0 with KOH). Anti-GFP antibody (0.4 mg/mL, affinity purified rabbit antibody) was flowed into the chamber and allowed to adhere non-specifically to the glass for 20 s. The surface was blocked using 0.5 mg/mL α-casein in buffer A. Excess protein was then washed away with buffer A. GFP-dynein motor domain was diluted in buffer A supplemented with 1 mM DTT and 0.1 mM ATP (2 mM DTT final; GFP-dynein 2 diluted to: 180 nM; GFP-dynein 1 diluted to: 60 nM), flowed into the chamber and allowed to bind for 20 s. Excess protein was washed away with buffer A. Reaction mix (25 mM PIPES, 90 mM KCl, 6 mM MgCl$_2$, 1 mM EGTA, 0.01% Triton X-100, 3 mM DTT, 20 µM taxol, 1 mM ATP, 1x oxygen scavenging system components [4.5 mg/mL glucose, 35 µg/mL catalase, 200 µg/mL glucose oxidase], 2% final DMSO [v/v] with compounds at appropriate concentrations, 0.05 mg/mL α-casein, microtubules [rhodamine labeled, purified from bovine brain, GMP-CPP and Taxol-stabilized], pH 7.0 with KOH) was flowed into the chamber using ~4 chamber volumes. For experiments shown in *Figure 4—figure supplement 1*, α-casein concentration in reaction mix was 0.5 mg/mL and all other parameters were held constant.

For reversibility experiments, motility in the presence of 5 µM **8** was observed as above, and upon completion of imaging, the chamber, fresh buffer A supplemented with 1 mM ATP was flowed into the chamber and incubated for 1 min; this wash and incubation cycle was repeated twice. Reaction mix containing DMSO (no compound) was flowed into the chamber using ~4 chamber volumes. The chamber was sealed with valap and returned to the microscope.

*Microscopy conditions:* Following addition of reaction mix, microtubules were allowed to bind for 5 min before imaging. For reversibility experiments, microtubules were allowed to bind for 10 min before imaging. Videos were taken on a Zeiss Axiovert 200M wide-field microscope equipped with a Zeiss 100x/1.45 NA α-Plan-Fluar objective. Data were captured with an EM-CCD camera (iXon DU-897, Andor Technology) with a 0.2 s exposure time and frame rate of 0.25–0.5/second. Four 10–20 s videos were captured per experimental condition. Velocities were measured using the kymograph tool in FIJI (*Schindelin et al., 2012*). Data analysis is discussed below under 'General data analysis procedures.'

## ATPase assays
### ATPase activity of dynein
ATPase assays using recombinant Dynein 1 (wild-type and mutants) were performed in a buffer containing 25 mM PIPES pH7.0, 30 mM KCl, 1 mM EGTA, 5 mM MgCl2, 0.01% triton X-100, 1 mM DTT, and 20–30 µM taxol (if microtubules were present). For typical conditions, enzyme was incubated

with varying concentrations of inhibitor in buffer containing 2% DMSO for 10 min in a volume of 10 µL. When microtubules were present, they were polymerized by stepwise addition of taxol and high concentrations of DMSO (30% final during polymerization reactions), and re-suspended following centrifugation in ATPase buffer to a concentration up to threefold above the final concentration used in the reaction (bovine brain tubulin, no fluoresecent label). For microtubule-stimulated reactions in dose-response format, reactions were assembled by 1:1:1 mixture of 3x enzyme stock, 3x compound stock (in buffer with 6% DMSO), and 3x microtubule stock to each well. For microtubule-stimulated ATPase assays at tubulin concentrations > 2.5 µM, enzyme stock was supplemented with compound stock in pure DMSO (resulting in an intermediate DMSO concentration of 6%), and then the microtubule stock was added. Final enzyme concentrations were 30–100 nM for GFP-dynein 1, wild-type 6x-His-dynein 1, and K2601A 6x-His-dynein 1 and up to 500 nM for K1912A 6x-His-dynein 1. The final DMSO concentration was 2% in all ATPase assays performed.

Assembled reaction mixtures (including enzyme, compound, and microtubules, if present) were incubated for 10 min before addition of ATP. Each ATPase reaction was initiated by addition of 2 µL of a 6x ATP stock containing trace $\gamma$-$^{32}$P ATP (3 µCi/mmol, 3 µCi/mL, Perkin Elmer) and allowed to proceed for a time predetermined to lie within the linear range of the assay (20–30 min). For typical conditions, ATP was used at a final concentration of 1 mM. Reactions were quenched by the addition of 12 µL 100 mM EDTA, and 2 µL of each quenched reaction was spotted onto PEI-cellulose TLC plates (Millipore). Plates were developed in a glass chamber with a freshly prepared solution of 150 mM formic acid and 150 mM LiCl, dried, exposed to a storage phosphor tray (GE Healthsciences), and scanned on a Typhoon imaging system (GE Healthsciences). The fraction of $\gamma$-phosphate hydrolyzed in each condition was quantified using FIJI.

### Ultraviolet light and ADP-vanadate-dependent photocleavage of dynein

Thirty-microliter reactions were prepared in 96-well plates as follows. ATPase buffer was supplemented with concentrated stocks of His-dynein 1 (90nM final), compound **8** (4µM final) or an equivalent amount of solvent carrier (3.3% DMSO), followed by a 10 min incubation at room temperature. ATP (0.1 mM final) was added and the mixtures were incubated 5 min at room temperature. NaVO$_3$ (0.1 mM final) was added and the mixtures were again incubated 5 min at room temperature. Vanadate was deliberately omitted from one well. The plate was then transferred to ice and irradiated with 365 nm light for 15 min with the lamp positioned 10 cm above the plate(Spectroline Maxima ML-3500S lamp). Following addition of SDS-PAGE loading buffer and heating (95°, 5 min), individual reactions were resolved on hand-poured 8% tris-glycine polyacrylamide gels. Gels were stained (Coomassie blue), imaged using a Li-Cor Odyssey, and band intensities were analyzed using FIJI.

### Assays of hedgehog pathway activity

NIH-3T3 cells stably expressing a luciferase reporter downstream of a Gli binding site (Shh-Light2 cells, RRID: CVCL_2721) were maintained in DMEM with 10% bovine calf serum (BCS). Cells were seeded at a density of 30,000 cells/well in 96-well tissue culture-treated plates (Corning, cat #353072) in 100 µL in DMEM +10% BCS, and incubated for 48 hr (*Taipale et al., 2000*). Wells were washed briefly with PBS. Next 100 µL of low serum media (DMEM +0.5% BCS) containing smoothened agonist (SAG, 500nM), and either solvent control (0.2% DMSO) or test compound (serial three-fold dilutions of each inhibitor starting from 20 µM) were added to the wells. After 28–32 hr of inhibitor treatment, cells were washed with 50 µL PBS and lysed for >30 min in 30 µL Passive Lysis Buffer (Promega Dual Luciferase kit, cat E1910). 5 µL of each lysate was transferred to white, solid-bottom 96-well plates (Greiner, cat #655075), followed by rapid addition (within 30 s) of 30 µL of Luciferase Assay Reagent using a multichannel pipette. Luminescence for each condition was read using a Synergy Neo plate reader (5 s integration time per well). Cell line identity was confirmed by measurement of the degree of response to Hedgehog pathway stimulation by the synthetic agonist SAG and using previously-published values from our groups and others as references (*Firestone et al., 2012*; *Hyman et al., 2009*).

### Cytotoxicity assays

Murine inner medullary collecting duct (IMCD3) cells stably expressing mNeonGreen-IFT88 (described previously [*Ye et al., 2013*]) were maintained in DMEM/F12 supplemented with 10% FBS.

Cell line identity was confirmed by the proper localization and motility of mNeonGreen IFT88 puncta, in reference to published analyses from our groups and from the work of others (*Ye et al., 2013*; *Yang et al., 2015*). Cells were seeded in a 96-well plate at a density of $5 \times 10^3$ cells/well. Following overnight incubation, low-serum media (0.2% FBS) containing solvent control (0.3% DMSO) or compound. At fixed time points, CellTiter-Glo 2.0 (Promega) reagent was added to the wells, mixed by pipetting, and transferred to a white, opaque 96-well plate. After a 10 min incubation, luminescence for each condition was read using a Synergy Neo plate reader (1 s integration time per well).

## Intraflagellar transport assays

Murine inner medullary collecting duct (IMCD3) cells stably expressing mNeonGreen-IFT88 were maintained in DMEM/F12 supplemented with 10% FBS. Cells were typically passaged fewer than 10 times following thawing of frozen stocks. Cells were described previously (*Ye et al., 2013*). $6 \times 10^5$ cells were seeded onto 22-mm coverslips in a six well plate and incubated for 24 hr at 37°C. The media was then replaced with DMEM/F12 +0.2% FBS and cells were incubated 24 hr to promote cilium formation. Immediately prior to imaging, media was replaced with phenol red-free media (Leibovitz's L-15 + 0.2% FBS) containing either carrier solvent control (0.3% DMSO) or compound. IFT88 transport was observed using a TE2000-E spinning disk confocal microscope (Nikon PlanApo 100x/1.45 objective lens) fitted with a Photometrics Cascade II (EMCCD 512) camera and imaged at a frame rate of 2 frames/s. Kymographs were generated in FIJI using *KymographClear* and the velocities of mNeonGreen-IFT88 foci movement were quantified by *KymographDirect* (*Mangeol et al., 2016*). The algorithm used to identify IFT88 foci could identify particles with velocities $\geq$ 25nm/s as moving particles.

For washout experiments, cells were exposed to a 5-min compound treatment as described above. Cells were then washed twice with 5 mL of L-15 supplemented with 10% FBSand 0.3% DMSO. A 5-min incubation followed each wash. Imaging was performed as described above immediately after the second 5-min wash and incubation step had been completed. For washout experiments shown in *Figure 4—figure supplement 3*, wash steps were performed with either L-15 supplemented with 10% FBS or L-15 supplemented with 0.2% FBS. Following washout of compound, cells were incubated at 37° for 60-min and then imaged.

## Lysosome motility assays

CAD cells (catecholaminergic neuronal tumor cell line, RRID:CVCL_0199) were maintained in DMEM/F12 medium with 10% Fetal Bovine Serum (*Qi et al., 1997*). Cell line identity was confirmed by the morphological response to serum withdrawal (i.e. neurite formation), a previously-documented characteristic of this cell line (*Qi et al., 1997*). 24 hr before the experiment cells were plated on coverslips in serum-free medium to induce neurite formation. Cells were treated with inhibitors or control solvent (0.1% DMSO final in all experiments) for 60 min, then lysosomes were labeled with 0.1 µM (final concentration) LysoTracker Red DND-99 (Molecular Probes) and used for microscopy immediately after LysoTracker addition.

Microscopy was performed using a Nikon TE-2000 inverted microscope equipped with a LED X-Cite illuminator (Lumen Dynamics) at 37 degrees. Images were acquired with a Hamamatsu ORCA-Flash 4.0 camera driven by MetaMorph software at a rate of 1 frame per second for 1 min. Lysosome motility in neurites was analyzed using Diatrack software (http://www.diatrack.org/), particle velocities were measured from frame to frame and the total length of trajectories divided by the number of analyzed organelles was calculated for every experimental condition.

## General data analysis procedures

Microtubule velocity measurements: the tracks of microtubules were traced and converted to kymographs (x axis-displacement, y axis-time) using the kymograph function of FIJI (RRID: SCR_002285). The angle formed between an edge of a microtubule and the horizontal was measured. This angle, θ, was converted into velocity using the following relationship: velocity = 1/|tangent(θ)| * (unit pixel distance)/(time interval between frames).

The calibrated pixel distance was 0.152 µm and the time interval between frames was either 2 or 4 s. The velocities of at least seven microtubules were measured per field of view, if sufficient tracks

could be found. In cases where fewer than seven microtubules were found in a field of view, velocities of all microtubules on the coverslip were measured. Approximately 30 microtubule velocities were measured for a given condition and their mean was reported as the average microtubule velocity.

Microtubule number counting: The number of microtubules bound to a coverslip in the first frame of a movie were counted manually.

Curve fitting: Where dose-response experiments are fit and $IC_{50}$ values given, these are obtained as follows: Individual experiments are treated as separate. The data are fit to a sigmoidal dose-response curve of the form (Y=Bottom + (Top-Bottom)/(1 + 10((LogIC50-X)*HillSlope)). All variables were allowed to float freely with the exception of the variable denoted 'bottom', which was constrained to be >0. The separate $IC_{50}$ values obtained from fits to each individual experiment's dataset were averaged together and presented as a mean with range (*Figure 3G and H*) or standard deviation (*Figures 3I*, *5D*, *6E and K*).

Statistical analysis of intraflagellar transport frequencies: Run frequencies output directly from *KymographDirect* were tabulated across all cilia imaged in a given condition (see *Figure 4—figure supplement 2*). For each of two experimental conditions (5 µM **8**, 10 µM **8**) and the respective controls, an un-paired T-test was performed for the difference of means control and compound treatment. This analysis was performed for each direction of motion.

Quantification of ADP-vanadate dependent photocleavage of dynein was performed as follows: Scanned gels were analyzed with FIJI, and fraction cleaved quantified as follows:

Fraction cleaved = (intensity of band at 250 kDa) /
(intensity of band at 250 kDa + intensity of band at ~350 kDa)

## Acknowledgements

We thank Professor Erwin Peterman and Dr. Pierre Mangeol for assistance with analysis of intraflagellar transport data. We acknowledge Mr. M Iida (Takeda Pharmaceuticals Company, Limited, Kanagawa, Japan) for valuable assistance with structural analysis of compounds. We thank Dr. Milica Tesic Mark and Dr. Henrik Molina (Rockefeller University) for assistance with mass spectrometry. This work was supported by the NIH (R01 GM098579 to TMK, R01 GM52111 to VIG, R01 GM113100 to JKC, and R01 GM089933 to MVN). TMK acknowledges the Robertson Therapeutic Development Fund for support. JBS was supported by NIH grant T32GM007739 to the Weill Cornell/Rockefeller/ Sloan-Kettering Tri-Institutional MD-PhD Program. RMM was supported by a Damon Runyon Cancer Research Foundation Postdoctoral Fellowship (DRG-2222–15). APC was supported by the Medical Research Council, UK (MC_UP_A025_1011) and a Wellcome Trust New Investigator Award (WT100387). The WCMC NMR facility was supported by NIH instrumentation grant S10 OD016320. The Proteomics Resource Center at The Rockefeller University was supported by Leona M and Harry B Helmsley Charitable Trust for mass spectrometer instrumentation.

## Additional information

### Funding

| Funder | Grant reference number | Author |
| --- | --- | --- |
| National Institutes of Health | R01 GM098579 | Tarun M Kapoor |
| Robertson Therapeutic Development Fund | | Tarun M Kapoor |
| Damon Runyon Cancer Research Foundation | DRG-2222-15 | Rand M Miller |
| Medical Research Council | MC_UP_A025_1011 | Andrew P Carter |
| National Institutes of Health | T32GM007739 | Jonathan B Steinman |
| National Institutes of Health | R01 GM52111 | Vladimir I Gelfand |
| National Institutes of Health | R01 GM113100 | James K Chen |
| Wellcome | WT100387 | Andrew P Carter |

National Institutes of Health R01 GM089933 Maxence V Nachury

The funders had no role in study design, data collection and interpretation, or the decision to submit the work for publication.

## Author contributions

JBS, Conceptualization, Supervision, Validation, Investigation, Visualization, Writing—original draft, Writing—review and editing; CCS, Formal analysis, Validation, Investigation, Visualization, Writing—review and editing, Performed and analyzed analyses of intraflagellar transport; RMM, Conceptualization, Validation, Investigation, Writing—original draft; LSY, Conceptualization, Validation, Investigation; ASS, Investigation, Visualization, Performed and analyzed lysosome motility experiments; HF, Conceptualization, Formal analysis, Supervision, Validation, Investigation, Writing—original draft, Designed and executed synthesis of compounds in this manuscript, Performed analytical validation of their identity and purity, Also supervised synthetic research efforts of others; SM, YT, MA, Formal analysis, Validation, Investigation, Designed and executed synthesis of compounds in this manuscript, Performed analytical validation of their identity and purity; MN, Formal analysis, Validation, Investigation, Performed and analyzed x-ray crystallography experiments; RZ, Resources, Supervision, Investigation; AEO, Formal analysis, Supervision, Investigation, Supervised and helped analyze two-dimensional NMR experiments; AGJ, Formal analysis, Investigation, Performed and helped analyze two-dimensional NMR experiments; FY, Supervision, Methodology, Writing—review and editing, Helped guide the analysis and interpretation of intraflagellar transport assays; MVN, Resources, Methodology, Writing—review and editing; YF, Supervision, Investigation, Led chemical synthesis efforts; KA, MAF, Supervision, Project administration; VIG, Supervision, Visualization, Writing—review and editing; JKC, Conceptualization, Supervision, Visualization, Writing—review and editing; APC, Resources, Supervision, Writing—review and editing; TMK, Conceptualization, Supervision, Validation, Investigation, Writing—original draft, Project administration, Writing—review and editing

## Author ORCIDs

Jonathan B Steinman, http://orcid.org/0000-0001-9492-4746
Tarun M Kapoor, http://orcid.org/0000-0003-0628-211X

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
