## [Decision Letter]

Thank you for submitting your article "Chemical structure-guided design of Dynapyazole, a potent cell-permeable dynein inhibitor with a unique mode of action" for consideration by *eLife*. Your article has been favorably evaluated by Philip Cole (Senior Editor) and three reviewers, one of whom, Wilfred van der Donk, is a member of our Board of Reviewing Editors.

The reviewers have discussed the reviews with one another and the Reviewing Editor has drafted this decision to help you prepare a revised submission.

Summary:

This study presents optimization of the chemical properties and potency of ciliobrevins for inhibiting of the microtubule (MT)-associated motor protein, cytoplasmic dynein. Based on conformational analysis of ciliobrevin, the authors covalently locked the preferred conformation in a series of analogs. The final result is an inhibitor, which the authors coin "dynapyrazole", with improved chemical properties and up-to 10-fold higher affinity. It appears to act differently than the parent compound in that it appears to be specific for the microtubule-induced activity and not the basal activity of dyneins. The amount of new biology learned appears modest and inconclusive, but if the essential revisions can be achieved, this work would benefit the field given the steadily increasing demand for understanding dynein function in cells and cilia. With good physiological analysis of dynapyrazole inhibition and reversibility, the paper could be very strong. Unless the questions in point 5 below can be addressed, the reviewers suggest this contribution to be a Tools and Resources paper with omission of the currently insufficiently supported mechanistic data.

Essential revisions:

1) Physiological analysis is limited to axonal transport of lysosomes. There is much potential information in these experiments, but much of this is currently lacking. The multicolor time projections of lysosomal movement are visually confusing. The reader will understand that the change from multicolored to white implies inhibition, but important details are missing. The lysosomal "displacements" are very long on average, and probably represent a composite of multiple +, –, and stalled behavior. The direction of movement is at minimum important as a test of dynapyrazole specificity – does it affect MT minus end -directed runs alone? Although the challenges to address this are acknowledged, if dynapyrazole is fast acting and physiologically specific, it should show initial dramatic minus end inhibition. At present, bidirectional inhibition is not addressed, but whether it occurs or not, and whether it sets in with increasing time of drug treatment, are important issues to clarify for the cell biological reader/user. Stalls are also important to quantify as evidence for coordinate +/- inhibition, but also as an indicator of cellular toxicity. Kymographs are a much better way to show detailed motile behavior and should be featured. The nature of inhibition can also be better defined for the community that would be using the new compound. Ciliobrevin is said to be a substrate competitor. How was this determined? Similar analysis should be performed for the new compound. Competition analysis between CB, dynapyrazole, and ATP might be useful. In order to show the reversible effect of dynapyrazole, washout experiments should be performed in a physiological setting e. g., lysosome transport.

2) At many places in the manuscript the statistical analyses is either not clear or not sufficient. Legends to Figure 2, Figure 3, Figure 4, Figure 5: For each measurement, the authors should list the number of data values they used to calculate the reported mean values and standard deviations. For example, in the legend to Figure 3, the authors write "ciliobrevin D: 20 μM (19-21 μM), n = 2", which could suggest that the authors performed one experiment at 19 μM ciliobrevin D and another at 21 μM. If so, how many microtubules were tracked for the reported velocities? Solely listing the number of replicates ("n") is not sufficient. Similarly, legend to Figure 3: If the second of two replicates was poorly fit (n = 2), why wasn't the experiment repeated more often to confirm that the second replicate was an "outlier"? Results, tenth paragraph: Also here, the authors should indicate whether the given "n = 7" corresponds to the number of replicates or the number of measured velocities. Results, ninth paragraph: In the cell-based assays, one cannot say that compound 4 inhibited the Hedgehog pathway with reduced potency compared to dynapyrazole since the two are within experimental error with the high error on the data for 4 (5.2 +- 5 μM).

3) Figure 4. The application of 5 μM dynapyrazole causes a reduction of the "run length" of lysosomes from ~32 μm to ~7 μm. While the ~32-μm runs can be seen in the color-coded image in Figure 4, the ~7 μm runs do not appear in the color-coded frames. While the authors write that the ~7 μm runs "can be seen as white punctate that result from the overlay of multiple color-coded frames without particle translocation", a significant fraction of the ~7 μm runs, which are only slightly shorter than the 10 μm scale bar Figure 4, should appear color-coded. It is therefore not clear to a reader why one cannot see the ~7 μm runs in the color-coded frames as colored. As mentioned above, kymographs are a better way to show detailed motile behavior.

4) The authors should address the following experimental concerns: The dynein preparations used for this analysis appear to be less than ideal. Single-headed recombinant dynein-1 and -2 motor domains are each used, though ATPase levels are only reported for dynein 1. However, the measured ATPase activity, especially in the presence of microtubules, is very low compared to native higher eukaryotic cytoplasmic dynein. This implies that a substantial fraction of the dynein may be inactive. The effects of ciliobrevin and dynapyrazole on dynein binding to MTs should also be evaluated and shown. It is possible that these compounds, or contaminants, might serve to oxidize dynein active site residues, and cause formation of dead-heads. MT gliding behavior also needs to be more thoroughly shown. The% of moving MTs and the distribution of velocities (histograms) should be provided.

5) The arguments that dynapyrazole targets the AAA1 site specifically are confusing and not well supported. This part of the study should be eliminated, unless it can be performed in a manner that provides conclusive insight in drug and motor mechanisms. If the authors want to keep their mechanistic interpretation, the following questions need to be addressed: Ciliobrevin effects on MT-stimulated ATPase activity should be included for proper comparison with dynpyrazole (Figure 5). The effects of dynapyrazole on the MT-activated ATPase activity of the AAA3 K/A mutant need to be analyzed. Also, if the effects of dynapyrazole on the basic ATPase activity of the AAA3 K/A mutant are mediated by AAA1, then the statement that dynapyrazole has only an effect of dynein's MT-activated ATPase activity is not correct. The authors should therefore present the effects of dynapyrazole on the MT-activated ATPase activity of the AAA3 K/A mutant and discuss the effects of dynapyrazole on the basal vs. MT-activated ATPase activity. The effect of blocking ATP binding to AAA3 on dynein's MT-activated ATPase activity should also be analyzed. While it is possible that the AAA3 K/A mutation abolishes the MT-activated ATPase activity of dynein, the authors should at least present and discuss this data. Also, if it is indeed microtubule-stimulated activity in the AAA1 site that is inhibited by dynapyrazole, would one not expect to see complete inactivation then in the AAA3 mutant? Figure 5 still seems to show incomplete inhibition. In addition, if AAA1 contributes only a small fraction of the basal activity, why can AAA1 activity then contribute significantly to the basal activity in the AAA3 K/A mutant? The authors don't discuss the possibility that AAA4 contributes significantly to the basal ATPase rate of dynein. The authors should at least discuss this possibility.

6) Based on the reduction in the velocity of dynein-based MT gliding as a result of the reduction in the MT-stimulated ATPase activity of dynein, the application of dynapyrazole is expected to cause a reduction in the velocity of lysosome movement. Why isn't the effect of dynapyrazole on the velocity of lysosome movement reported? It would be highly interesting if differential effects on transport distance and velocity were found.

---

## [Author Response]

Essential revisions:

1) Physiological analysis is limited to axonal transport of lysosomes. There is much potential information in these experiments, but much of this is currently lacking. The multicolor time projections of lysosomal movement are visually confusing. The reader will understand that the change from multicolored to white implies inhibition, but important details are missing. The lysosomal "displacements" are very long on average, and probably represent a composite of multiple +, -, and stalled behavior. The direction of movement is at minimum important as a test of dynapyrazole specificity – does it affect MT minus end -directed runs alone? Although the challenges to address this are acknowledged, if dynapyrazole is fast acting and physiologically specific, it should show initial dramatic minus end inhibition. At present, bidirectional inhibition is not addressed, but whether it occurs or not, and whether it sets in with increasing time of drug treatment, are important issues to clarify for the cell biological reader/user. Stalls are also important to quantify as evidence for coordinate +/- inhibition, but also as an indicator of cellular toxicity. Kymographs are a much better way to show detailed motile behavior and should be featured. The nature of inhibition can also be better defined for the community that would be using the new compound. Ciliobrevin is said to be a substrate competitor. How was this determined? Similar analysis should be performed for the new compound. Competition analysis between CB, dynapyrazole, and ATP might be useful. In order to show the reversible effect of dynapyrazole, washout experiments should be performed in a physiological setting e. g., lysosome transport.

We thank the reviewers for raising all these different points. The changes we have made to address these concerns are:

A) Physiologic analysis: The physiologic analyses in the original submission included inhibition of lysosome motility and Hedgehog signaling. To address your concern, we have now obtained cell lines from the group of Dr. Max Nachury and with their guidance have done live-cell imaging of intraflagellar transport in living cells. To obtain the level of detailed analysis you requested (velocity distributions, frequencies) we used computer-based tracking algorithms. The findings are presented in a new figure (Figure 4) in the revised manuscript.

The benefit of studying the effect of dynapyrazole-A on intraflagellar transport is that a comparable study using temperature sensitive alleles of dynein 2 was available, which provided us a valuable reference point for the activity of our compound. Briefly, we find that inhibition of intraflagellar transport by dynapyrazole at a concentration near its IC_50_ for dynein 2-driven microtubule gliding is very similar to that observed in a previous study of the effects of temperature-sensitive mutants of dynein 2 on intraflagellar transport in *Chlamydomonas* (Engel et al., J. Cell Biol. 2012, 199 (1): 151-167 DOI: 10.1083/jcb.201206068, [Figure 2]). Specifically, the authors noted that a ~60% reduction in retrograde velocity upon transfer to the restrictive temperature was accompanied by a modest (~10-20% reduction) in anterograde velocity, values that closely match those we observed. Furthermore, particle frequencies were reduced by 30-60% in both directions, a change that mirrors the effect we observed upon dynapyrazole-A treatment. Based on these data we believe that dynapyrazole will be a valuable tool to examine how dynein-2 driven transport within the cilium is linked to its dynamic structure and functions.

B) Lysosome motility: As requested, we have added new kymograph analysis of lysosome motility. With these analyses, we now show how the distribution of lysosome velocities is affected by dynapyrazole-A treatment. The kymographs reveal bidirectional inhibition of lysosome movement following treatment with dynapyrazole-A. This finding is consistent with other analyses of organelle transport that indicate a biochemical and/or mechanical coupling between different motors on a single cargo. Multiple groups have studied the coupling between retrograde- and anterograde-directed motion of organelles in cells, tissue preparations, and living organisms and have consistently observed that depletion or inhibition of dynein motors leads to reduction in bidirectional motion (Gross et al., J. Cell Biol. 2002, 156 (4): 715, DOI: 10.1083/jcb.200109047, Martin et al., Mol. Biol. Cell, 1999 Vol. 10, 3717, doi: 10.1091/mbc.10.11.3717, Waterman-Storer et al., PNAS, 1997 Vol 94, No. 22, 12180, Ally et al., J. Cell Biol. 2009, 187 (7): 1071, DOI: 10.1083/jcb.200908075). We further note that the ciliobrevins caused bi-directional reduction in peroxisome motion (Firestone et al.,Nature 2012, Apr. Vol 484, 125-129. DOI: doi:10.1038/nature10936 [Figure 4]). Additional analyses, at fast temporal resolution to uncover a transient or "initial dramatic minus end inhibition" would be interesting, but we hope that you agree this result is outside the scope of this current manuscript that now includes additional physiological analyses, i.e. analyses of intraflagellar transport.

To address your concern that this bidirectional inhibition may reflect general toxicity on this time scale, we have also examined if intracellular ATP concentrations are reduced under the conditions of the lysosome imaging experiments. ATP levels are unchanged over 3 hours at dynapyrazole-A concentrations (5µM) that inhibit lysosome transport. We observed toxicity at 3 hours at higher dynapyrazole-A concentrations (15µM), but we cannot exclude that this toxicity is a result of inhibition of dynein, which has roles in many vital cellular processes. These data are included in the revised manuscript and should help others with the use of these inhibitors in cellular contexts.

C) Inhibitor reversibility: As per your request, we have now examined reversibility of dynapyrazole-A treatment in cells. We observed full reversal of inhibition of retrograde intraflagellar transport following washout of dynapyrazole-A. We note that the reversibility was more efficient if we included serum in the cell culture medium, consistent with the hydrophobic nature of these compounds. The text clearly describes these observations.

D) Competition with nucleotide binding. You also raise the question about competition with ATP and in response, to further characterize the mode of inhibition by dynapyrazole-A in vitro, we now include new data analyzing ADP-vanadate-dependent photocleavage of dynein. We now show that dynein photocleavage, which has been shown to occur at AAA1, is inhibited by dynapyrazole-A treatment, consistent with inhibition of nucleotide binding at AAA1 of dynein. We note that the concentration of dynapyrazole-A necessary to block photocleavage of dynein (4µM) is ~40 times lower than was reported for ciliobrevin A (Firestone et al.,Nature 2012, Apr. Vol 484, 125-129. DOI: doi:10.1038/nature10936 [supplementary figure 15]).

We believe that with these additions, we have both improved the analysis of the physiologic activity of dynapyrazole-A and strengthened our mechanistic analysis of how this compound inhibits dynein. It is our sincere hope that the reviewers will agree.

2) At many places in the manuscript the statistical analyses is either not clear or not sufficient. Legends to Figure 2, Figure 3, Figure 4, Figure 5: For each measurement, the authors should list the number of data values they used to calculate the reported mean values and standard deviations. For example, in the legend to Figure 3, the authors write "ciliobrevin D: 20 μM (19-21 μM), n = 2", which could suggest that the authors performed one experiment at 19 μM ciliobrevin D and another at 21 μM. If so, how many microtubules were tracked for the reported velocities? Solely listing the number of replicates ("n") is not sufficient. Similarly, legend to Figure 3: If the second of two replicates was poorly fit (n = 2), why wasn't the experiment repeated more often to confirm that the second replicate was an "outlier"? Results, tenth paragraph: Also here, the authors should indicate whether the given "n = 7" corresponds to the number of replicates or the number of measured velocities. Results, ninth paragraph: In the cell-based assays, one cannot say that compound 4 inhibited the Hedgehog pathway with reduced potency compared to dynapyrazole since the two are within experimental error with the high error on the data for 4 (5.2 +- 5 μM).

We thank the reviewers for their careful attention to our data.

We have added the number of microtubules whose velocities we quantified and have clarified the replicate number at several locations. For example:

Figure 2. Number of microtubules quantified: DMSO-327, Cil. D 85, 2-98, 3-90, 4-77.Figure 3. Number of microtubules quantified: 8: 10µM-36, 5µM-59, 2.5µM-98, 1.3µM-112, 0.6µM-102, 0.3µM-126; G-Ciliobrevin D: 80µM-10, 40µM-47, 20µM-78, 10µM-85, 5µM-99, 2.5µM-66, 1.3µM-80;Figure 3. Number of microtubules quantified: 6: 20µM-38, 8µM-24, 3.2µM-48, 1.3µM-50, 0.5µM-53; H-9: 20µM-29, 8µM-54, 3.2µM-50, 1.3µM-54, 0.5µM-56.

Please note that we have added additional measurements to our analysis of the effect of cyclization (Figure 3).

We have also added the following sentence to the figure legends of Figure 3, Figure 5 and Figure 6 to clarify the meaning of IC_50_ values reported: "IC_50_ values reported reflect the mean (with range or S.D.) of separate IC_50_ values obtained from independent dose-response analyses." We acknowledge that our initial legends were confusing and hope this sentence provides clarity. In particular, the numbers 19 and 21 µM denote the range between two independent measurements of the IC_50_ of dynein 2-driven gliding by ciliobrevin D that allowed us to obtain the average IC_50_ value.

3) Figure 4. The application of 5 μM dynapyrazole causes a reduction of the "run length" of lysosomes from ~32 μm to ~7 μm. While the ~32-μm runs can be seen in the color-coded image in Figure 4, the ~7 μm runs do not appear in the color-coded frames. While the authors write that the ~7 μm runs "can be seen as white punctate that result from the overlay of multiple color-coded frames without particle translocation", a significant fraction of the ~7 μm runs, which are only slightly shorter than the 10 μm scale bar Figure 4, should appear color-coded. It is therefore not clear to a reader why one cannot see the ~7 μm runs in the color-coded frames as colored. As mentioned above, kymographs are a better way to show detailed motile behavior.

We acknowledge the reviewers' concerns regarding the manner in which we have presented our data. We agree with the reviewers’ suggestion that a kymograph is a more suitable way to display these data and have made this change. In the new analysis we have provided of lysosome motility, we have added representative kymographs and velocity distribution histograms. We also wish to clarify that the 7µm run value reported at 5µM dynapyrazole-A reflects the displacement of only those particles that did move, not a mean displacement value for all particles. Our data show that, at the 5µM dynapyrazole-A dose, ~60% of particles have a velocity of <25 nm/s in both retrograde and anterograde directions.

4) The authors should address the following experimental concerns: The dynein preparations used for this analysis appear to be less than ideal. Single-headed recombinant dynein-1 and -2 motor domains are each used, though ATPase levels are only reported for dynein 1. However, the measured ATPase activity, especially in the presence of microtubules, is very low compared to native higher eukaryotic cytoplasmic dynein. This implies that a substantial fraction of the dynein may be inactive. The effects of ciliobrevin and dynapyrazole on dynein binding to MTs should also be evaluated and shown. It is possible that these compounds, or contaminants, might serve to oxidize dynein active site residues, and cause formation of dead-heads. MT gliding behavior also needs to be more thoroughly shown. The% of moving MTs and the distribution of velocities (histograms) should be provided.

We appreciate the reviewers' concerns regarding the protein activity we report. Respectfully, we believe that the confusion about protein hydrolysis activity may arise from the common practice of presenting the ATPase activity of dynein in units of nmol/min/mg protein. When converted to these units (molecular weight of the construct is ~400kDa), the basal ATPase activity of our single-headed dynein 1 construct is ~90nmol/min/mg protein (~0.6s^-1^), which is in line with the basal ATPase rates reported by others for both native and recombinant mammalian dyneins (Shpetner et al., bovine dynein: ~40 nmol/mg/min, J Cell Biol. 1988 Sep;107(3):1001-9. [Figure 1A]; Nicholas et al., rat dynein ~6 s^-1^ (per dimer), Nat Commun. 2015 Feb 11;6:6206 [Supplementary Figures 2A, 2B]; Ori-McKenney et al., mouse dynein, ~50 nmol/mg/min, Nat Cell Biol. 2010 Dec;12(12):1228-34 [Figure 1C]). We note that the ATPase activity observed here is lower than those observed in dyneins from *S. cerevisiae* and, especially, from *D. Discoideum*. We observe modest stimulation (~2 fold) at 2.5µM microtubules, and have now added data showing that we observe up to 5-fold stimulation of ATPase activity at higher microtubule concentrations (20µM), which is within the 2- to 6-fold range of stimulation observed at high microtubule concentrations for rat, cow, and mouse dynein 1 constructs (Ori-McKenney et al., Figure 1C, Shpetner et al., Figure 1A, Nicholas et al., Supplementary Figure 2B). Therefore, we believe, and hope you agree, that the basal and microtubule-stimulated activities we report are in line with the range of activities reported by others for mammalian dyneins.

In addition, to address your concerns, we have added a deeper analysis of microtubule velocity. As requested, we now show histograms showing microtubule velocity in the DMSO controls as well as in the presence of a range of concentrations of dynapyrazole-A. We note that the fraction of stopped or very slow microtubules is low for both dynein isoforms tested.

We have also added an analysis of the effect of dynapyrazole-A on microtubule number in the dynein-driven microtubule gliding assays, which can be used as a readout of dynein-microtubule affinity. We observe that the effect of dynapyrazole-A on coverslip microtubule number is different for each dynein isoform: a marked reduction is noted for dynein 2, while a slight increase is noted for dynein 1. As dynapyrazole-A disrupts nucleotide binding to AAA1, which itself is critical for regulating dynein-microtubule affinity, it is not surprising that compound treatment affects this affinity. Dissecting the differences that lead to a divergent effect of dynapyrazole on the dynein 2-microtubule interaction compared to the dynein 1-microtubule interaction will require careful biophysical analysis that we hope you agree is outside the scope of this study. As we have noted in the manuscript, we focused on our detailed biochemical analyses on dynein 1, the better characterized isoform.

5) The arguments that dynapyrazole targets the AAA1 site specifically are confusing and not well supported. This part of the study should be eliminated, unless it can be performed in a manner that provides conclusive insight in drug and motor mechanisms. If the authors want to keep their mechanistic interpretation, the following questions need to be addressed: Ciliobrevin effects on MT-stimulated ATPase activity should be included for proper comparison with dynpyrazole (Figure 5). The effects of dynapyrazole on the MT-activated ATPase activity of the AAA3 K/A mutant need to be analyzed. Also, if the effects of dynapyrazole on the basic ATPase activity of the AAA3 K/A mutant are mediated by AAA1, then the statement that dynapyrazole has only an effect of dynein's MT-activated ATPase activity is not correct. The authors should therefore present the effects of dynapyrazole on the MT-activated ATPase activity of the AAA3 K/A mutant and discuss the effects of dynapyrazole on the basal vs. MT-activated ATPase activity. The effect of blocking ATP binding to AAA3 on dynein's MT-activated ATPase activity should also be analyzed. While it is possible that the AAA3 K/A mutation abolishes the MT-activated ATPase activity of dynein, the authors should at least present and discuss this data. Also, if it is indeed microtubule-stimulated activity in the AAA1 site that is inhibited by dynapyrazole, would one not expect to see complete inactivation then in the AAA3 mutant? Figure 5 still seems to show incomplete inhibition. In addition, if AAA1 contributes only a small fraction of the basal activity, why can AAA1 activity then contribute significantly to the basal activity in the AAA3 K/A mutant? The authors don't discuss the possibility that AAA4 contributes significantly to the basal ATPase rate of dynein. The authors should at least discuss this possibility.

We thank the reviewers for their constructive suggestions regarding our mechanistic analysis. We now include analysis of ADP-vanadate-dependent photocleavage of dynein, which sheds light on the mechanism dynein inhibition by dynapyrazole-A. We show that dynein photocleavage, which has been shown to occur at AAA1, is inhibited by dynapyrazole-A treatment, consistent with inhibition of nucleotide binding at AAA1 of dynein. The concentration of dynapyrazole-A necessary to block photocleavage of dynein (4µM) is ~40 times lower than was reported for ciliobrevin A (Firestone et al).

We have added many of the other analyses requested. We now show the effect of ciliobrevin D on the microtubule-stimulated ATPase activity of dynein, which reveals inhibition to ~40% residual activity at the highest concentration tested (40µM), with no saturation of the inhibitory effect observed. We compare the potencies of inhibition of the basal ATPase activity of the AAA3 mutant of dynein by ciliobrevin D and dynapyrazole-A and show that the new compound is ~8-fold more potent in this setting. We show that the AAA3 mutant of dynein is stimulated by microtubules, but to a lesser degree than observed for the wild-type enzyme, consistent with analyses of comparable mutants by others (Kon et al., Biochemistry. 2004 Sep 7;43(35):11266-74, [Table 1]). We show that dynapyrazole-A suppresses microtubule stimulation of both the wild-type and AAA3 mutant enzymes across a broader range of microtubule concentrations.

The new data we present support our previous assertion that dynapyrazole-A primarily blocks the microtubule-stimulated component of dynein’s ATPase activity while ciliobrevin D inhibits the basal and microtubule-stimulated activities of dynein. When taken together with the inhibition of ADP-vanadate-dependent photocleavage at AAA1 of dynein by low concentrations of dynapyrazole-A, we believe that our data now show more clearly that (A) dynapyrazole-A blocks ATP hydrolysis at AAA1, (B) dynapyrazole-A only weakly inhibits basal ATPase activity, (C) dynapyrazole-A strongly inhibits microtubule-stimulated hydrolysis by dynein. We believe these observations are best explained by a model in which AAA1 contributes minimally to the basal ATPase activity of dynein and contributes significantly (and in a microtubule concentration-dependent manner) to the microtubule-stimulated activity of dynein. In this model, inhibition by dynapyrazole-A of AAA1 only would be expected to give results consistent with those we observe. We have attempted to clarify the writing describing our conclusions.

The reviewers asked: “if AAA1 contributes only a small fraction of the basal activity, why can AAA1 activity then contribute significantly to the basal activity in the AAA3 K/A mutant?” In current models nucleotide occupancy within one ATPase sites of dynein regulates that at other sites. Thus, in the AAA3 mutant, this site is not only impaired in its own nucleotide binding and hydrolysis, but also its APO-like nucleotide state may be sensed and transmitted to other AAA sites, leading to their activation. We note in the manuscript that others have observed comparable and seemingly counterintuitive *stimulation* of dynein’s basal rate in constructs with a similar AAA3 walker A lysine-to-alanine mutation. We have added the following statement to the manuscript in the attempt to clarify our reasoning. “Our finding that the AAA3 mutant has elevated basal activity relative to the wild-type is consistent with the protein being in a state that mimics one in which hydrolysis at AAA1 is activated.”

We appreciate the reviewers’ suggestion that another site (likely AAA4) might also contribute hydrolysis activity in the setting of the AAA3 mutant. Our newly-added microtubule stimulation data for the AAA3 mutant further support this conclusion, and we have revised the text accordingly.

6) Based on the reduction in the velocity of dynein-based MT gliding as a result of the reduction in the MT-stimulated ATPase activity of dynein, the application of dynapyrazole is expected to cause a reduction in the velocity of lysosome movement. Why isn't the effect of dynapyrazole on the velocity of lysosome movement reported? It would be highly interesting if differential effects on transport distance and velocity were found.

We appreciate the reviewers' enthusiasm for the potential for dynapyrazole to illuminate the roles of dynein-driven motion in a cellular context. As mentioned above, we have included an expanded analysis of lysosome motility as well as a new analysis of intraflagellar transport. We demonstrate a reduction in the velocities of moving lysosomes in addition to the track shortening previously shown. We also now show that dynapyrazole causes reversible inhibition of intraflagellar transport.